# Rethinking Hebbian Principle: Low-Dimensional Structural Projection for Unsupervised Learning

**Shikuang Deng**[1*]**, Jiayuan Zhang**[2*]**, Yuhang Wu**[1]**, Ting Chen**[1]**, Shi Gu**[3,4✉]

[1]School of Computer Science and Engineering, UESTC
[2]Glasgow College, UESTC
[3] School of Computer Science and Technology, Zhejiang University
[4] State Key Lab of Brain-Machine Intelligence, Zhejiang University, China
dengsk@uestc.edu.cn, gus@zju.edu.cn

## Abstract

Hebbian learning is a biological principle that intuitively describes how neurons adapt their connections through repeated stimuli. However, when applied to machine learning, it suffers serious issues due to the unconstrained updates of the connections and the lack of accounting for feedback mediation. Such shortcomings limit its effective scaling to complex network architectures and tasks. To this end, here we introduce the Structural Projection Hebbian Representation (SPHeRe), a novel unsupervised learning method that integrates orthogonality and structural information preservation through a local auxiliary nonlinear block. The loss for structural information preservation backpropagates to the input through an auxiliary lightweight projection that conceptually serves as feedback mediation while the orthogonality constraints account for the boundedness of updating magnitude. Extensive experimental results show that SPHeRe achieves SOTA performance among unsupervised synaptic plasticity approaches on standard image classification benchmarks, including CIFAR-10, CIFAR-100, and Tiny-ImageNet. Furthermore, the method exhibits strong effectiveness in continual learning and transfer learning scenarios, and image reconstruction tasks show the robustness and generalizability of the extracted features. This work demonstrates the competitiveness and potential of Hebbian unsupervised learning rules within modern deep learning frameworks, demonstrating the possibility of efficient and biologically inspired learning algorithms without the strong dependence on strict backpropagation. Our code is available at https://github.com/brain-intelligence-lab/SPHeRe.

## 1 Introduction

In recent years, deep learning based on error backpropagation has achieved revolutionary developments in the field of artificial intelligence [1, 2]. However, the biological plausibility of backpropagation remains a subject of debate, particularly concerning its requirement for symmetric feedback connections and precise global error signal transmission, mechanisms not readily observed in biological neural circuits [3, 4]. At the same time, some neuroanatomical studies have found that the plasticity regulation of cortical synapses mainly follows local rules, with the strength changes dynamically dominated by the activities of pre-/postsynaptic neurons, occasionally under the synergistic regulation of higher-order signals such as neuromodulators. Based on these neuroanatomical studies, researchers have proposed various biologically inspired learning algorithms to replace the backpropagation algorithm. The Hebbian rule, as a canonical algorithm of this type, was first pro-

---

* Equal contribution
✉ Corresponding author

39th Conference on Neural Information Processing Systems (NeurIPS 2025).

posed by Donald Hebb in 1949 [5]. Its core idea is that "neurons that fire together wire together," which means that learning and memory formation are achieved through the cooperative strengthening of synapses. As it does not rely on global signals, it belongs to unsupervised learning.

In order to overcome the limitations of the original Hebbian rules in terms of stability, competitiveness, and feature extraction capabilities, researchers have developed various important variants based on them, such as threshold mechanisms, Bienenstock-Cooper-Munro (BCM) rules [6], and Oja's rules [7]. A significant theoretical leap was made by Pehlevan, Chklovskii, and collaborators, who established a principled connection between Hebbian learning and similarity-matching objectives [8, 9, 10]. Their foundational work demonstrated that optimizing objectives of the form $||X^\top X - Y^\top Y||_F^2$—which aim to preserve the geometric structure (Gram matrix) of the input—can be achieved through fully local Hebbian and anti-Hebbian synaptic updates. However, extending these elegant theories to deep, nonlinear networks often theoretically relies on feedback connections for inter-layer credit assignment. More recent works, such as combining Oja's rule with Winner-Take-All (WTA) mechanisms [11, 12], have achieved impressive results on complex datasets, yet many Hebbian-inspired methods still lack a clear, optimizable objective that seamlessly integrates with modern deep learning paradigms, hindering their scalability.

In this paper, we revisit this problem from a different architectural perspective. We propose the Structural Projection Hebbian Representation (SPHeRe), a novel, Hebbian-inspired, greedy layer-wise unsupervised learning framework. We begin by simplifying the equivalent loss of Oja's rule, arriving at a stable objective that, coincidentally, aligns with the similarity-matching goals pioneered by Pehlevan and Chklovskii.Then we provide theoretical proof, demonstrating that minimizing this simplified loss corresponds to finding an optimal low-dimensional projection of the input data (Lemma 4.2.1), akin to principal subspace analysis (PCA). Our core innovation lies in how we apply this objective. Instead of deriving synaptic-level rules or relying on feedback, we introduce a lightweight auxiliary projection module ($\phi$), creating a purely feedforward, block-wise training architecture. This "main-pathway learning, side-pathway supervision" design decouples the main network's high-dimensional feature learning from the low-dimensional structural preservation objective. This allows the main block ($f$) to learn rich representations ($Y'$) while the auxiliary block ($\phi$) efficiently computes the structural loss on a low-dimensional projection ($Z$). Complemented by an orthogonality constraint ($L_{orth}$) to encourage feature decorrelation, SPHeRe provides a practical and effective bridge between Hebbian principles and modern deep learning. Our experiments show that SPHeRe achieves SOTA performance among Hebbian-inspired unsupervised methods and demonstrates strong generalization in continual and transfer learning, positioning it as a competitive greedy layer-wise pre-training strategy. The following summarizes our main contributions:

- We propose SPHeRe, a novel Hebbian-inspired unsupervised learning framework, whose core innovation is a purely feedforward, block-wise training architecture. By introducing a lightweight auxiliary module ($\phi$), SPHeRe decouples high-dimensional feature learning in the main pathway from a low-dimensional structural preservation objective, enabling the effective application of Hebbian principles to deep, non-linear networks.

- We provide a theoretical analysis (Lemma 4.2.1 and Appendix B) showing that optimizing the core loss $\mathcal{L}_{\text{SPHeRe}}$ is equivalent to finding the optimal low-dimensional projection (principal subspace) of the input data. Through experiments on reconstruction and transfer learning, we further verify that SPHeRe can extract robust and generalizable image features.

- We demonstrate through extensive experiments that SPHeRe achieves state-of-the-art performance among Hebbian-inspired unsupervised methods on CIFAR-10/100 and Tiny-ImageNet. Furthermore, its strong performance in continual learning, transfer learning, and feature reconstruction validates its effectiveness as a greedy layer-wise pre-training strategy that learns robust and generalizable representations.

## 2    Related Work

**Synaptic Plasticity Rules.**    Synaptic plasticity, the modulation of connection strength between neurons, is recognized as a fundamental biological substrate for learning, memory consolidation, and behavioral shaping [13]. The classical Hebbian principle, which posits that correlated activity strengthens synapses [5], laid the foundation for understanding activity-dependent modifications. But neuroscience experiments reveal that the rules of synaptic plasticity in the brain are much more

complex, such as the timing-dependent effects of STDP [14], network-wide balancing through homeostatic plasticity [15], influences between synapses through heterosynaptic plasticity [16], and even the control of plasticity itself, termed metaplasticity [17]. Recognizing the power of these biological adaptation strategies, researchers are increasingly working to integrate similar neuroplastic concepts into neural networks. Notable examples include investigations into the functional impact of homeostatic regulation in artificial systems [15], the synergistic use of Hebbian and non-Hebbian rules within recurrent architectures [18], and the development of unsupervised learning paradigms based on complementary Hebbian and anti-Hebbian updates [12], demonstrating a growing trend towards biologically inspired learning mechanisms in modern artificial intelligence. A significant advancement in connecting Hebbian learning to principled optimization was made by [8, 9]. They demonstrated that optimizing a similarity-matching objective, which minimizes the discrepancy between input and output Gram matrices, can be mathematically decomposed into a set of purely local, online synaptic update rules with Hebbian and anti-Hebbian forms. Subsequent work extended this framework to deep networks [10], but these extensions often theoretically rely on feedback connections to propagate error or target signals for credit assignment across layers. Starting from Oja's rule, we obtain the same core mathematical objective as they did. But we propose a fundamentally different architectural solution that is a purely feedforward, block-based approach applying this goal to deep networks and avoiding the need for explicit feedback paths.

**Unsupervised learning.** Unsupervised learning (UL) aims to learn data representations from unlabeled data. Classical methods such as K-means [19] and Principal Component Analysis (PCA) [20] exploit the statistical structure of the data to perform dimensionality reduction or clustering. In the context of image classification, recent advances in UL for neural networks are typically categorized into two main approaches: self-supervised learning (SSL) and the reconstruction method [21]. SSL methods, including contrastive learning approaches such as SimCLR [22], BYOL [23], and SimSiam [24], rely on pretext tasks and often employ pseudo-labels. The reconstruction method, on the other hand, trains the encoder and decoder simultaneously through image reconstruction loss to train the encoder to extract low-dimensional representations of the images, and the decoder can reconstruct the images from the low-dimensional representations. The autoencoder [25], VAE [26], and MAE [27] are representative of this type of method. However, the inherent limitations of these mainstream SSL and reconstruction methods are that they fundamentally rely on end-to-end gradient computation through backpropagation, a process that lacks biological plausibility. In this paper, we attempt to establish a biologically inspired new path for unsupervised learning.

## 3   Preliminaries

### 3.1   Hebbian Rule

The Hebbian learning rule satisfies the locality property, where the synaptic connection is determined solely by the spike states of its pre- and post-synaptic neurons, without the influence of other neurons. In the classical Hebbian rule, the equation is $\Delta w_{ij} = \eta \cdot v_i v_j$, representing the product of the firing rates of the pre- and post-synaptic neurons. Then consider a simple linear matrix multiplication process, we use $X \in \mathbb{R}^{B \times N}$ to represent the input vector, $Y \in \mathbb{R}^{B \times M}$ to represent the output vector, $W \in \mathbb{R}^{N \times M}$ to represent the synaptic weights, and the relationship between them is: $Y = X \cdot W$. So, the original Hebbian rule for $W$ is:

$$\Delta W = \eta \cdot (X^\top Y), \tag{1}$$

### 3.2   Oja's rule

The classical Hebbian rule has certain deficiencies, such as the potential for overtraining and instability due to consistently positive firing rates. The Oja's rule is a well-known variant of the Hebbian rule, incorporating an additional decay term in its formula; the greater the post-synaptic activity, the more significant the decay. Under Oja's rule, the weight update equation is $\Delta w_{ij} = \eta(v_i - w_{ij}v_j)v_j$. And the Oja's rule formula for $W$ can be expressed as follows:

$$\Delta W = \eta(X^\top \cdot Y - W \cdot Y^\top \cdot Y). \tag{2}$$

To some extent, Oja's rule approximates principal component analysis (PCA) for dimensionality reduction and has the effect of extracting features. However, the current Oja's rule does not consider

the complex nonlinear transformations and feature quantity adjustments in deep learning, making it difficult to directly apply to multilayer networks.

# 4 Methodology

This section introduces the Structural Projection Hebbian Representation (SPHeRe) method, an unsupervised learning approach inspired by Hebbian principles but adapted for modern deep learning architectures. SPHeRe aims to learn meaningful representations by preserving the structural relationships within the input data, utilizing a specific loss function, an orthogonality constraint, and a lightweight auxiliary block.

## 4.1 Analyze Hebbian rules from the perspective of loss functions

In this section, we will analyze how the Hebbian rule can extract information from the perspective of the loss function. Since the Hebbian rule does not care for its activation function, we just start from the simplest linear layer, where we use $X \in \mathbb{R}^{B \times N}$ to represent the input vector, $Y \in \mathbb{R}^{B \times M}$ to represent the output vector, $W \in \mathbb{R}^{N \times M}$ to represent the synaptic weights, and the relationship between them is: $Y = X \cdot W$.

**Original Hebbian rule.** For the simple linear layer trained using the classical Hebbian rule, its equivalent loss function is $\mathcal{L}_{\text{hebb}} = -\frac{1}{2}||Y||_F^2$. Minimizing the loss $\mathcal{L}_{\text{hebb}}$ will increase the overall magnitude of the output matrix $Y$, which means that the values of the elements in $Y$ will increase. This optimization might lead to a larger variance in the distribution of $Y$, making it easier to classify. However, without constraints, this could result in excessively large output values, potentially causing numerical instability. In contrast, the equivalent loss function of anti-hebbian is $\mathcal{L}_{\text{anti-hebb}} = \frac{1}{2}||Y||_F^2$, and optimizing $\mathcal{L}_{\text{anti-hebb}}$ is equivalent to driving all elements of the output $Y$ as close to zero as possible. This is a form of L2 regularization for the output.

**Oja's rule.** Oja's rule improves upon the classical Hebbian rule by introducing a suppressive term that prevents the weight matrix from growing indefinitely. The equivalent loss function of Oja's rule is:

$$\mathcal{L}_{\text{oja}} = \frac{1}{4}\text{Tr}\left((YY^\top - XX^\top)(XX^\top)^{-1}(YY^\top - XX^\top)\right). \qquad (3)$$

In which Tr denotes the trace of a matrix. Optimizing this loss function is somewhat equivalent to minimizing the Gram matrix (sample relationship matrix) $K_X = XX^\top$ of the input $X$ and the gram matrix $K_Y = YY^\top$ of the output $Y$. When $M \geq N$, the weight $W$ tends to $WW^\top \approx I_N$; when $M < N$, the output $Y$ becomes a representation of $X$, which will mimic the geometric structure of $X$ in the N-dimensional space as much as possible in the M-dimensional space, thus preserving the pairwise relationships between samples as much as possible. In the loss function, $(XX^\top)^{-1}$ acts as a weighting mechanism, weighting the errors according to the characteristics of the input data. However, the existence of $(XX^\top)^{-1}$ requires $(XX^\top)$ to be invertible, and it may lead to numerical instability.

## 4.2 Structural Projection Hebbian Representation (SPHeRe)

### 4.2.1 Preserving Data Structure

SPHeRe starts from the equivalent loss function of Oja's rule. First, we simplify the loss function by removing the inverse term $(XX^\top)^{-1}$ and the constant term. This new objective directly seeks to learn an output representation $(Y)$ that preserves the pairwise structural information inherent in the input, as captured by their respective Gram matrices ($XX^\top$ and $YY^\top$). While derived from a Hebbian learning rule (Oja's), this simplified loss also connects to dimensionality reduction techniques aiming for optimal data projection. We thus define the core SPHeRe loss function as:

$$\mathcal{L}_{\text{SPHeRe}} = \text{Tr}\left((YY^\top - XX^\top)(YY^\top - XX^\top)\right) = \left\|YY^\top - XX^\top\right\|_F^2. \qquad (4)$$

From the above equation, it can be intuitively seen that the goal of simplifying the loss function is to make the output Gram matrix approach the input Gram matrix. Some works use this method to achieve multidimensional scaling (MDS) [28] or non-negative matrix factorization (NMF) [29].

Notably, while we derive this objective from a Hebbian perspective, it is mathematically identical to the similarity-matching objectives studied extensively by [8]. The theoretical justification for this loss in capturing essential data structure comes from its connection to principal subspace projection, as formalized in the following lemma.

**Lemma 4.2.1.** *Let $X \in \mathbb{R}^{B \times N}$ be the input data matrix and let $Y = XW \in \mathbb{R}^{B \times M}$ be the output, $W \in \mathbb{R}^{N \times M}$ be the weight matrix, and $M < N$. Consider the loss function $\mathcal{L} = \|YY^\top - XX^\top\|_F^2$. Minimizing $\mathcal{L}$ with respect to $W$ yields an optimal output $Y^*$ given by $Y^* = XV_M$, where $V_M$ consists of the first $M$ right singular vectors of $X$. This $Y^*$ represents the projection of the rows of $X$ onto the $M$-dimensional principal subspace, and $Y^*Y^{*\top}$ is the best rank-$M$ approximation of $XX^\top$ in the Frobenius norm.*

The relevant proof is provided in Appendix A. From the lemma 4.2.1, it can be concluded that the objective of the new equivalent loss function is to find the optimal projection of the low-dimensional space of the input $X$, and the minimum value of the equivalent loss function is $\sum_{i=M+1}^{N} \sigma_i^4$, where $\sigma_i$ is the $i$-th largest singular value of the input $X$. In addition, when the relationship between $Y$ and $X$ is nonlinear, such as $Y = f(X)$, where $f$ is a nonlinear neural network (universal approximator). Due to the powerful fit ability of $f$, minimizing the loss function $\mathcal{L}$ will still lead $Y$ to approach the optimal solution $Y^* = XV_M$ to minimize $\mathcal{L}$.

After obtaining the equivalent loss function, we can find the derivative of $W$ as: $\nabla_W = 4X^\top(YY^\top - XX^\top)Y$. The outer term $X^\top(\cdot)Y$ still reflects the fundamental Hebbian rule: the change in weights is related to the correlation between pre-synaptic activity ($X$) and post-synaptic activity ($Y$). $(YY^\top - XX^\top)$ is the weight term of the fundamental Hebbian rule. It wants to transform the statistical structure of the input space ($XX^\top$) into the statistical structure of the output space ($YY^\top$) and adjust the Hebbian rule through the higher-order relationships between the sample pairs in the input and output.

### 4.2.2 Enhancing Features: Orthogonality constraint

To enhance the quality and interpretability of the extracted features, we introduce an additional Orthogonality constraint term, called orthogonal loss. This loss encourages the different features represented by the columns of the output matrix $Y$ to be mutually orthogonal. Decorrelated or orthogonal features often lead to less redundant representations and can improve the effectiveness of downstream tasks. The Orthogonal Loss function is defined as:

$$\mathcal{L}_{\text{orth}} = \left\|Y^\top Y - I_M\right\|_F^2. \tag{5}$$

The matrix $Y^\top Y \in \mathbb{R}^{M \times M}$ computes the product of dots between all pairs of columns (features) of $Y$. So, minimizing $\mathcal{L}_{\text{orth}}$ pushes the columns of $Y$ towards orthogonality. By promoting orthogonality among the features, we aim to reduce redundancy in the learned representation, ensuring that each feature captures distinct aspects of the input data. Under orthogonal loss, the gradient of $W$ is: $\nabla_W = X^\top Y(Y^\top Y - I_M)$. Interestingly, the gradient associated with this loss also exhibits a Hebbian modulation structure, which adds a weight term $Y^\top Y - I_M$ to the standard Hebbian rule and drives the output features of $Y$ to be as orthogonal as possible.

As a result, the total loss combines the new hebbian loss and the orthogonality loss and can be described as:

$$\mathcal{L}_{\text{Total}} = \mathcal{L}_{\text{SPHeRe}} + \lambda\mathcal{L}_{\text{orth}}, \tag{6}$$

where $\lambda$ is a hyperparameter that balances the contribution of the orthogonality constraint against the low-rank projection.

### 4.2.3 Integrating SPHeRe into Neural Networks: The Auxiliary block

The simplified Hebbian loss function $\mathcal{L}_{\text{SPHeRe}}$ and the orthogonal loss $\mathcal{L}_{\text{orth}}$ presented earlier are defined based on a linear transformation $Y = XW$. However, deep learning heavily relies on non-linear layers to capture complex data patterns. Directly applying linear Hebbian loss to train deep neural networks is often insufficient. Therefore, we extend our Hebbian method to non-linear settings.

**Linear to nonlinear.** we replace the linear transformation with a nonlinear mapping represented by a neural network. Let $Y' = f(X, W)$ denote the output of a neural network layer (or block) $f$ with parameters $W$, given the input $X$. According to Lemma 4.2.1, minimizing the simplified Hebbian loss $\mathcal{L}_{\text{SPHeRe}}$ drives the output $Y$ towards the optimal low-dimensional projection of the input $X$. Meanwhile, the neural network $f$ is a universal approximator, it has the capacity to learn complex functions, including the optimal linear projection identified by Lemma 4.2.1 if that minimizes the objective. Thus, even with a non-linear function $f$, optimizing the new Hebbian loss that compares input and output statistical structures can guide $f$ to learn meaningful representations. We provide the corresponding verification experiments in Appendix B.

**Auxiliary block.** To further enhance flexibility and computational efficiency, we introduce an additional lightweight non-linear projection layer $\phi$ with parameters $\theta$. This layer takes the intermediate representation $Y'$ as input and produces the auxiliary output $Z = \phi(Y', \theta) = \phi(f(X, W), \theta)$. We provide a comparison between SPHeRe with the auxiliary block and other similar methods in Appendix G. This two-stage non-linear mapping offers several advantages:

- **Computational Efficiency.** The dimension $M_Z$ of the auxiliary output $Z$ can be chosen to be small (e.g., $M_Z \ll M'$), significantly reducing the cost of computing $ZZ^\top$ and the $\mathcal{L}_{\text{SPHeRe}}$ loss, making it practical for large batches and deep networks.

- **Flexibility for Main Block ($f$).** The main block $f$ can learn a potentially high-dimensional ($M'$) and complex intermediate representation $Y'$ without being directly constrained by the low-rank structure imposed by $\mathcal{L}_{\text{SPHeRe}}$. The auxiliary network $\phi$ handles the projection to the low-dimensional space $Z$ where the structural comparison occurs.

- **Effective Nonlinear Learning.** Even though $f$ and $\phi$ are nonlinear, minimizing $\mathcal{L}_{\text{SPHeRe}}$ on $Z$ still effectively guides the network. As indicated by Lemma 4.2.1, the target for $Z$ is the principal subspace projection $XV_{M_Z}$. Since $f$ and $\phi$ are universal approximators, they possess the capacity to learn the complex mapping from $X$ to an approximation of $XV_{M_Z}$ through $Y'$ and $Z$ during optimization. Experimental validation (detailed in Appendix B) confirms this: using this non-linear auxiliary setup, the learned auxiliary features $Z$ exhibit high structural similarity (measured by Centered Kernel Alignment (CKA) and alignment of Singular Value Decomposition (SVD) components) to features obtained from the ideal linear projection defined by Lemma 4.2.1. This demonstrates that $\mathcal{L}_{\text{SPHeRe}}$ effectively drives the non-linear system to capture the principal structural information of the input via the auxiliary branch.

- **Locality.** The objective of $W$ is determined locally by comparing its input $X$ to a representation $Z$ derived (via $\phi$) from its output $Y'$. Although gradients still flow through $f$ and $\phi$ during optimization, the objective itself is defined relative to the block's input and (projected) output structure, differing from end-to-end supervision from a final task loss.

## 4.3 Distinguishing SPHeRe from PCA and Oja's Rule

While inspired by principles related to Oja's rule and PCA , SPHeRe differs significantly as a practical deep learning method: (1) **Nonlinear:** SPHeRe is explicitly designed for training deep, nonlinear neural networks layer-by-layer or block-by-block, whereas standard PCA is a linear technique and Oja's rule is typically analyzed in linear or simple non-linear settings; (2) **Auxiliary block:** The use of the auxiliary block $\phi$ is a core architectural innovation of SPHeRe. It decouples the dimensionality required for efficient structural comparison ($M_Z$) from the dimensionality of the main block feature representation ($M'$), allowing flexibility and scalability not present in direct PCA/Oja implementations; (3) **Motivation** The primary goal is unsupervised pre-training or layer-wise training of deep networks to learn features beneficial for downstream tasks (like classification, transfer learning), rather than solely dimensionality reduction as in standard PCA. We provide a comparison of SPHeRe with more unsupervised dimensionality reduction methods in Appendix G.

## 4.4 Overall Method

The core idea of SPHeRe is to leverage an auxiliary lightweight nonlinear projection $\phi$ to compute a low-dimensional representation $Z$, whose relationship structure (captured by a kernel matrix $K_Z$) is driven to match the relationship structure of the input $X$ (captured by $K_X$). This is achieved

by minimizing the simplified Hebbian loss $\mathcal{L}_{\text{SPHeRe}}$ defined on these kernel matrices. Then, an orthogonality constraint $\mathcal{L}_{\text{orth}}$ can be applied to the output $Y'$ of the main block $f$ to encourage the decorrelation of the features. The auxiliary branch allows the main block $f$ to operate with potentially higher-dimensional intermediate features $Y'$ while keeping the Hebbian loss computation efficient via the low-dimensional $Z$. The process can be summarized by the following equations and Fig. 1:

$$Y = f(X, W), \quad Z = \phi(Y, \theta)$$
$$\hat{X} = X/\|X\|_2, \quad \hat{Z} = Z/\|Z\|_2$$
$$(K_X) = XX^\top, \quad (K_Z) = ZZ^\top$$
$$\mathcal{L}_{\text{SPHeRe}} = \|K_Z - K_X\|_F^2, \quad \mathcal{L}_{\text{orth}} = \|Z^\top Z - I\|_F^2$$
$$\mathcal{L}_{\text{Total}} = \mathcal{L}_{\text{SPHeRe}} + \lambda\mathcal{L}_{\text{orth}}$$

Figure 1: The concept of SPHeRe.

# 5 Experiments

## 5.1 Experimental Setup

Our Hebbian network architecture consists of three Hebbian-trained convolutional layers (384, 768, 1536 channels) with kernel size 3 and a fully connected output layer trained with backpropagation. Each convolutional layer is followed by a max-pooling layer to reduce the image resolution by half. A skip connection is employed in the final layer, in which an Avgpool(2,2) is used to downsample, but gradients are not propagated backward through it. If not specifically stated, we use Leaky-ReLU as the activation function. A lightweight auxiliary network, $\phi$, is introduced for the downsampling of convolutional features. Let the input feature map of $\phi$ have $k$ channels. The network $\phi$ comprises three sequential stages: (i) a $1 \times 1$ convolution halving the channel dimension to $k/2$; (ii) an Adaptive Pooling operation collapsing the spatial dimensions to $1 \times 1$; (iii) a fully connected layer projecting the intermediate $k/2$-dimensional features onto a final 256-dimensional representation. We trained the network using AdamW (learning rate: 0.001, weight decay: 0.05) with a batch size of 128 and a learning rate scheduler; the hyperparameter $\lambda$ for $\mathcal{L}_{\text{orth}}$ is set to 0.8; no data augmentation was applied except for standard normalization. For fair comparison, the SoftHebb baseline adopted the same network structure but otherwise retained its standard optimal hyperparameters.

## 5.2 Comparison to Existing Works

In this section, we compare our method with other existing unsupervised synaptic plasticity learning methods on the CIFAR-10, CIFAR-100 and Tiny-ImageNet datasets. For the SoftHebb method, we use open source code and verify it in our network architecture. It is possible that SoftHebb is sensitive to network structure, which has led to a little decrease in network performance than reported in the paper after using our structure. As shown in Table 1, our method consistently outperforms these existing methods on all three benchmark datasets. On CIFAR-10, our method achieves the test accuracy of 81.11%, exceeding the best previous result (80.3%). This advantage is maintained on CIFAR-100 (0.79% higher than SoftHebb) and becomes even more pronounced on the more complex Tiny-ImageNet dataset, where our method achieves 40.33% compared to 34.12% for SoftHebb.

Table 1: Compare with existing synaptic plasticity works. * denotes results reported on the source paper. † denotes self-implemented result on our network structure with open source code. All accuracy values are presented in the format mean ± standard deviation.

| Approaches | CIFAR-10 | CIFAR-100 | Tiny-ImageNet |
|---|---|---|---|
| D-CSNN [30] | 73.7 | 45.17 | 14.36 |
| Hard WTA [31] | 72.2 | 32.56 | – |
| Hard WTA [11] | 74.6 | – | – |
| SoftHebb [12]* | 80.3 | 56.0 | – |
| SoftHebb [12]† | 78.86 | 54.18 | 34.12 |
| **SPHeRe** | **81.11 ± 0.11** | **56.79 ± 0.69** | **40.33 ± 0.24** |

## 5.3 Analysis Experiments

In this section, we conduct a series of analysis experiments to evaluate different aspects of our proposed method. Due to space limitations in the main text, we present a selection of key results here. For a more comprehensive evaluation and additional analysis experiments, we refer the reader to Appendix B,H,I,J,K,L, which provides more analysis experiments and results.

### 5.3.1 Ablation Study

In this section, we conduct an ablation study on different components of our method to verify their effectiveness to the final performance. Specifically, we decompose our method into the original Oja's Hebbian loss $\mathcal{L}_{oja}$, the simplified Oja's Hebbian loss $\mathcal{L}_{SPHeRe}$, the orthogonality constraint loss of features $\mathcal{L}_{orth}$, and the auxiliary block $\phi$. Then, we verify the unsupervised classification performance on CIFAR-10 with different combinations. The results are summarized in Table 2. Using only $\mathcal{L}_{oja}$ can achieve an accuracy of 73.8%, indicating that the original Oja's rule is still effective. It is worth noting that ablation experiments show that both the orthogonality constraint loss $\mathcal{L}_{orth}$ and the auxiliary block $\phi$ can effectively improve the algorithm's performance. However, $\mathcal{L}_{orth}$ requires a significant amount of computation, leading to memory overflow when dimension reduction is not performed using $\phi$. Although $\mathcal{L}_{SPHeRe}$ does not perform as well as the original $\mathcal{L}_{oja}$, its combination with other components ($\mathcal{L}_{orth}$ and $\phi$) is superior to the original Oja loss $\mathcal{L}_{oja}$. Ultimately, with the combination of $\mathcal{L}_{SPHeRe}$, $\mathcal{L}_{orth}$, and $\phi$, the network achieved 81.18% accuracy on the test set.

Table 2: Accuracy comparison for different loss function combinations

| $\mathcal{L}_{Oja}$ | $\mathcal{L}_{SPHeRe}$ | $\mathcal{L}_{orth}$ | $\phi$ | **Accuracy (%)** |
|---|---|---|---|---|
| ✓ | | | | 73.8 |
| ✓ | | | ✓ | 76.2 |
| ✓ | | ✓ | ✓ | 76.3 |
| | ✓ | | | 65.4 |
| | ✓ | | ✓ | 78.9 |
| | | ✓ | ✓ | 80.7 |
| | ✓ | ✓ | ✓ | **81.18** |

Table 3: Continual Learning results. * denotes self-implemented. All accuracy values are presented in the format mean ± standard deviation.

| Approaches | Split-CIFAR100 | Split-TinyImageNet |
|---|---|---|
| EWC [32] | 71.96± 0.37 | 62.87± 0.31 |
| HAT [33] | 75.70± 0.50 | 54.97± 0.84 |
| OWM [34] | 70.89± 0.13 | – |
| GPM [35] | 74.99 ± 0.12 | 66.00 ± 0.24 |
| SoftHebb [12]* | 51.1 | – |
| **SPHeRe** | 72.72± 1.14 | 63.46± 0.79 |
| **SPHeRe-EWC** | **76.53± 0.64** | **67.05± 1.16** |

### 5.3.2 Performance on Continual Learning

Continual learning (CL) is a significant challenge in machine learning, which requires the network to learn new knowledge sequentially without suffering catastrophic forgetting of previously acquired knowledge. We evaluated our proposed method, SPHeRe, on standard CL benchmarks, Split-CIFAR100 and Split-TinyImageNet, with the results presented in Table 3. Our approach achieves competitive performance, demonstrating its potential in the CL setting. The reason for this phenomenon may be that our unsupervised learning method pays more attention to the general representations of input images, so it is more robust to distribution changes encountered between tasks, thus mitigating the phenomenon of knowledge forgetting. Furthermore, when combined with a well-established CL technique (EWC), the SPHeRe-EWC method significantly improves performance, achieving state-of-the-art results on both datasets.

## 5.4 Performance on Transfer Learning

Since our method belongs to unsupervised learning, it can capture some general representations of the underlying data of images, which may be applicable to different target tasks or datasets. In this section, we conduct cross-dataset transfer learning experiments, training the three convolutional layers on one dataset (source) through SPHeRe and only training the classification head on another dataset (target) to evaluate network performance. The results, detailed in Table 4, show promising transferability. When transferring knowledge from Tiny-ImageNet to CIFAR-10, the model achieved a test accuracy of 80.03%, which is remarkably close to the 81.11% accuracy obtained when training SPHeRe directly on CIFAR-10 (a performance gap of only -1.08%). Similarly, transferring from CIFAR-10 to Tiny-ImageNet yielded a test accuracy of 37.7%, compared to the 40.33% baseline achieved with direct training on Tiny-ImageNet (a gap of -2.63%). Although there is a slight decrease in performance compared to training directly on the target dataset, these results indicate that the feature extraction rules learned by SPHeRe on the source dataset remain effective when applied to different target datasets, confirming the method's effective transfer learning capability.

Table 4: Cross-Dataset transfer learning performance of SPHeRe

| Transfer Direction | Transfer Learning (%) (Train/Test) | Non-Transfer (%) (SPHeRe alone) | Gap (vs. Non-Transfer) |
|---|---|---|---|
| Tiny-ImageNet $\rightarrow$ CIFAR-10 | 97.3 / 80.03 | 81.11 | $-1.08\%$ |
| CIFAR-10 $\rightarrow$ Tiny-ImageNet | 99.9 / 37.7 | 40.33 | $-2.63\%$ |

### 5.4.1 Reconstruction Experiments

In continuous learning and transfer learning experiments, SPHeRe achieved good experimental results, indicating that SPHeRe tends to extract more general information from images. To verify this statement, we designed an experiment to prove whether the image information extracted by SPHeRe can better compress the complete information of the image. We first pretrained both the baseline model and our SPHeRe model on the CIFAR-10 dataset. These pretrained models were then used as encoders to transform input images into feature maps. These images could be added with Gaussian noise. Next, we send the feature maps into a decoder, which was trained via backpropagation to reconstruct the original images. More details on the structure, training settings, and more results can be found in the Appendix H. Reconstruction loss (pixel-wise mean square error), calculated as the difference between the original and decoded images, served as our evaluation metric. As shown in Fig. 2, for reconstructing original clean images, the decoder utilizing features from our SPHeRe encoder achieved a remarkably low loss of $3.37 \times 10^{-3}$. This outperforms the reconstruction based on features of other methods. At the same time, visually, our method provides the best image restoration, indicating that our method can better compress and retain the complete image information. Furthermore, even under these noisy conditions, the decoder using SPHeRe features maintained an outstanding reconstruction performance, achieving an MSE of just $3.59 \times 10^{-3}$, effectively reconstructing the original clean image from the noisy input's features. Additionally, we notice that the SoftHebb method tends to discard the color information of the image. This may lead to decreased performance in tasks where color information is crucial. These reconstruction results strongly support our hypothesis that SPHeRe's unsupervised Hebbian learning mechanism captures more fundamental, complete, and robust visual information compared to both standard backpropagation and the SoftHebb baseline.

## 6 Conclusion

This work revisits Hebbian learning's connection to optimal low-dimensional projection via equivalent loss functions. We introduce SPHeRe, a novel unsupervised loss function derived by simplifying the equivalent loss of Oja's rule, designed to minimize structural differences between input and output representations. To enhance feature extraction, SPHeRe incorporates lightweight nonlinear modules and a feature orthogonality constraint. SPHeRe achieves state-of-the-art results among unsupervised plasticity methods on multiple classification datasets and demonstrates robustness and generalizability across continual learning, transfer learning, and image reconstruction tasks. Our findings highlight the feasibility and potential of Hebbian learning in modern deep learning.

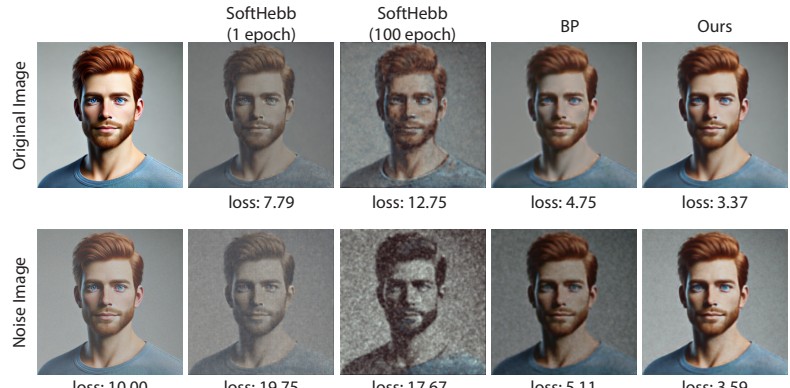

Figure 2: A sample of reconstruction results. The loss values are scaled by the factor of $10^3$.

However, certain limitations invite further investigation and delineate promising avenues for future work. One notable observation from our experiments is that performance gains slightly diminish as the number of convolutional layers increases. This suggests that noise accumulates as the layer increases and that purely local unsupervised SPHeRe learning lacks sufficient contextual signals to learn to eliminate noise. This challenge resonates with biological reality, where neural circuits employ a sophisticated synergy between local synaptic plasticity and global neuromodulatory signals. SPHeRe currently emphasizes the former. Therefore, a compelling next step involves exploring the integration of multiscale regulatory or contextual signals, perhaps inspired by biological reward/punishment pathways or attention mechanisms. We believe that pursuing such multiscale approaches will not only advance the capabilities and scalability of Hebbian and other biologically inspired algorithms within modern deep learning frameworks but also offer valuable computational perspectives that could deepen our understanding of biological neural learning processes.

# 7 Acknowledgment

This project is supported by NSFC Key Program (No. 62236009), Youth Program (No. 62506065), Postdoctoral Fellowship Program of CPSF under Grant Number GZC20251045.

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

# A  Proof of Lemma 4.2.1

**Lemma 4.2.1.** *Let $X \in \mathbb{R}^{B \times N}$ be the input data matrix and let $Y = XW \in \mathbb{R}^{B \times M}$ be the output, $W \in \mathbb{R}^{N \times M}$ be the weight matrix, and $M < N$. Consider the loss function $\mathcal{L} = \|YY^\top - XX^\top\|_F^2$. Minimizing $\mathcal{L}$ with respect to $W$ yields an optimal output $Y^*$ given by $Y^* = XV_M$, where $V_M$ consists of the first $M$ right singular vectors of $X$. This $Y^*$ represents the projection of the rows of $X$ onto the $M$-dimensional principal subspace, and $Y^* Y^{*\top}$ is the best rank-$M$ approximation of $XX^\top$ in the Frobenius norm.*

The proof of the above theorem is similar to the proof of the Eckart-Young-Mirsky theorem, and its proof is given below:

*Proof.* The objective is to minimize the loss function: $\mathcal{L}(W) = \|YY^\top - XX^\top\|_F^2$. Substituting $Y = XW$, we get:

$$\mathcal{L}(W) = \|(XW)(XW)^\top - XX^\top\|_F^2$$
$$= \|XWW^\top X^\top - XX^\top\|_F^2$$

Let $P = WW^\top$. The matrix $P \in \mathbb{R}^{N \times N}$ is semidefinite positive symmetric (SPD). The rank of $P$ is bounded by the dimensions of $W$: $\text{rank}(P) = \text{rank}(W) \leq \min(N, M) = M$, since $M < N$. The optimization problem can be reformulated to find an SPD matrix P that minimizes

$$\mathcal{L}(P) = \|XPX^\top - XX^\top\|_F^2. \tag{7}$$

First, we can perform a Singular Value Decomposition (SVD) of the input matrix $X$ that let

$$X = U\Sigma V^\top, \tag{8}$$

where:

- $U \in \mathbb{R}^{B \times B}$ is an orthogonal matrix ($U^\top U = I_B$).

- $V \in \mathbb{R}^{N \times N}$ is an orthogonal matrix ($V^\top V = I_N$). The columns $v_1, v_2, \ldots, v_N$ of $V$ are the right singular vectors of $X$.

- $\Sigma \in \mathbb{R}^{B \times N}$ is a rectangular diagonal matrix containing the singular values $\sigma_1 \geq \sigma_2 \geq \cdots \geq \sigma_r > 0$ on its main diagonal, where $r = \text{rank}(X)$.

We can express the terms in Eqn. 7 using the Eqn. 8:

- $XX^\top = (U\Sigma V^\top)(U\Sigma V^\top)^\top = U\Sigma V^\top V \Sigma^\top U^\top = U(\Sigma\Sigma^\top)U^\top$. Let $\Lambda = \Sigma\Sigma^\top \in \mathbb{R}^{B \times B}$. $\Lambda$ is a diagonal matrix with diagonal entries $\sigma_1^2, \ldots, \sigma_r^2, 0, \ldots, 0$.

- $XPX^\top = (U\Sigma V^\top)P(U\Sigma V^\top)^\top = U\Sigma V^\top PV \Sigma^\top U^\top$.

Then Eqn. 7 is change to:

$$\mathcal{L}(P) = \|U\Sigma V^\top PV \Sigma^\top U^\top - U\Lambda U^\top\|_F^2$$
$$= \frac{1}{4}\|\Sigma V^\top PV \Sigma^\top - \Lambda\|_F^2,$$

where the second equals sign is due to the unitary invariance property of the Frobenius norm (i.e., $\|QAZ\|_F = \|A\|_F$ for orthogonal matrices $Q, Z$). Let $Q = V^\top PV$. Since $V$ is orthogonal and $P$ is symmetric SPD with $\text{rank}(P) \leq M$, $Q$ is also symmetric PSD with $\text{rank}(Q) = \text{rank}(P) \leq M$. The optimization problem transforms into finding a symmetric PSD matrix $Q \in \mathbb{R}^{N \times N}$ with $\text{rank}(Q) \leq M$ that minimizes:

$$\mathcal{L}(Q) = \|\Sigma Q\Sigma^\top - \Lambda\|_F^2. \tag{9}$$

Let $\Sigma_r = \text{diag}(\sigma_1, \ldots, \sigma_r)$. We can appropriately partition $\Sigma$ and $Q$. Without loss of generality regarding the dimensions $B, N$, the product takes the form:

$$\Sigma Q\Sigma^\top = \begin{pmatrix} \Sigma_r & 0 \\ 0 & 0 \end{pmatrix}_{B \times N} Q \begin{pmatrix} \Sigma_r & 0 \\ 0 & 0 \end{pmatrix}_{N \times B}^\top = \begin{pmatrix} \Sigma_r Q_{11} \Sigma_r & 0 \\ 0 & 0 \end{pmatrix}_{B \times B}, \tag{10}$$

where $Q_{11}$ is the top-left $r \times r$ principal submatrix of $Q$. The matrix $\Lambda = \Sigma\Sigma^\top = \begin{pmatrix} \Sigma_r^2 & 0 \\ 0 & 0 \end{pmatrix}_{B \times B}$.

The Eqn. 9 can be simplified to:

$$\mathcal{L}(Q) = \|\Sigma_r Q_{11}\Sigma_r - \Sigma_r^2\|_F^2. \tag{11}$$

Let $A_{11} = \Sigma_r Q_{11}\Sigma_r$. Since $\sigma_i > 0$ for $i = 1, \ldots, r$, $\Sigma_r$ is invertible. Thus, $Q_{11} = \Sigma_r^{-1}A_{11}\Sigma_r^{-1}$. $A_{11}$ must be symmetric SPD and $\text{rank}(A_{11}) = \text{rank}(Q_{11}) \leq M$. The problem is equivalent to finding the best rank-$M$ symmetric SPD approximation $A_{11}$ to the diagonal matrix $\Sigma_r^2 = \text{diag}(\sigma_1^2, \ldots, \sigma_r^2)$ in the Frobenius norm. And the optimal rank-$k$ approximation of a diagonal matrix (in Frobenius norm) is obtained by keeping the $k$ diagonal entries with the largest magnitudes and setting the others to zero. Since $\sigma_1^2 \geq \cdots \geq \sigma_r^2 > 0$ and $A_{11}$ must be SPD, the best rank-$M$ symmetric SPD approximation $A_{11}^*$ to $\Sigma_r^2$ is obtained by keeping the first $M$ largest diagonal entries (assuming $r \geq M$, otherwise keep all $r$): $A_{11}^* = \text{diag}(\sigma_1^2, \ldots, \sigma_M^2, 0, \ldots, 0)_{r \times r}$. The corresponding optimal $Q_{11}^*$ is: $Q_{11}^* = \Sigma_r^{-1}A_{11}^*\Sigma_r^{-1} = \text{diag}(1, \ldots, 1, 0, \ldots, 0)_{r \times r}$, where there are $M$ ones. The optimal $N \times N$ matrix $Q^*$ can be constructed as: $Q^* = \begin{pmatrix} Q_{11}^* & 0 \\ 0 & 0 \end{pmatrix}_{N \times N} = \begin{pmatrix} I_M & 0 \\ 0 & 0 \end{pmatrix}_{N \times N}$. $Q^*$ is symmetric SPD and $\text{rank}(Q^*) = M$. Then, we have

$$\begin{aligned} P^* &= V\begin{pmatrix} I_M & 0 \\ 0 & 0 \end{pmatrix} N \times N V^\top \\ &= [V_M, VN - M]\begin{pmatrix} I_M & 0 \\ 0 & 0 \end{pmatrix}\begin{pmatrix} V_M^\top \\ V_{N-M}^\top \end{pmatrix} \\ &= V_M I_M V_M^\top = V_M V_M^\top \end{aligned}$$

$P^*$ is the projection matrix onto the subspace spanned by the first $M$ right singular vectors of $X$. We need to find $W^* \in \mathbb{R}^{N \times M}$ such that $W^*(W^*)^\top = P^* = V_M V_M^\top$. A valid solution is $W^* = V_M$. (Other solutions $W^* = V_M O$ for any $M \times M$ orthogonal matrix $O$ exist, but they lead to the same $YY^\top$). As a result, the optimal output $Y^*$:

$$Y^* = XW^* = XV_M = U\Sigma V^\top V_M = U\Sigma_{B \times M} = U_M\Sigma_M, \tag{12}$$

So the optimal output $Y^* = XV_M$ calculates the coordinates of the rows of $X$ projected onto this principal subspace. Furthermore, the product $Y^*(Y^*)^\top$ is: $Y^*(Y^*)^\top = (U_M\Sigma_M)(U_M\Sigma_M)^\top = \sum_{i=1}^M \sigma_i^2 u_i u_i^\top$, which is the best rank-$M$ approximation of $XX^\top = U\Lambda U^\top = \sum_{i=1}^r \sigma_i^2 u_i u_i^\top$ in the Frobenius norm. And the minimal loss $\mathcal{L}_{min} = \sum_{i=M+1}^N \sigma_i^4$.

$\square$

## B  SPHeRe Loss in Nonliner Condition

In practice, instead of directly using the linear projection, we applied a nonlinear transformation to $X$, we use $f$ to represent the nonlinear transformation. So the loss function is change to:

$$\mathcal{L} = \|f(XW; \theta_f)f(XW; \theta_f)^\top - XX^\top\|_F^2, \tag{13}$$

where the parameters $\theta_f$ of a universal approximator $f$ (neural network). When $Z = f(XW; \theta_f) \in \mathbb{R}^{B \times M'}(M' < N)$, through Lemma 4.2.1, we can obtain the optimal $Z^* = XV_{M'}$ that minimizes the loss $\mathcal{L}$. Since $f$ is a universal approximator, there theoretically exists a set of parameters $\theta_f^*$ and $W^*$ such that $f(XW^*; \theta_f^*) = XV_{M'}$. Therefore, during the training process, the output $Z$ of the auxiliary branch will gradually approach the dimensionality reduction projection of $X$: $XV_{M'}$, and the output $Y$ of the main branch layer will gradually summarize and extract the features of the input $X$ through this training process.

To verify our viewpoint, we conducted the corresponding experiments. We trained two different network branches: 1) linear branch, where $f$ is a linear mapping with parameter $W$; 2) nonlinear branch, where $f$ is a neural network with 3 layers. Both branches receive the exact same input data $X$, and we train them through Eqn. 13. During the training process, we record the output features at different training epochs and use the centered kernel alignment (CKA) similarity to quantify and compare the similarity between the output features between two different branches (Fig. 3 A). From

the CKA similarity curve in Fig. 3 B, it can be seen that, after only 20 training epochs, the CKA similarity exceeded 0.90 and tended to stabilize at a high level as the training progressed for all datasets. This indicates that regardless of whether an explicit nonlinear transformation $f$ is used, the SPHeRe loss drives the network to learn very similar feature representations.

Secondly, to further explore the structure of the feature space, we performed a single value decomposition (SVD) on the feature output of the two branches. We focus on the 36 main components (that is, right singular vectors) and calculate the similarity between the feature vectors obtained by linear SPHeRe optimization (Linear $f$) and those obtained by nonlinear SPHeRe optimization (Nonlinear $f$). We visualized these similarities as a $36 \times 36$ similarity matrix (Fig. 3 C). The diagonal region of the similarity matrix shows very high similarity, especially for the first few principal components with lower indices (the first 20). This indicates that the most important principal directions (directions explaining the most variance) in the feature space learned by the two optimization methods are highly aligned and consistent. Principal components with higher indices (the last 10) do not strictly follow the maximum similarity on the diagonal, but most singular vectors have relatively similar corresponding vectors.

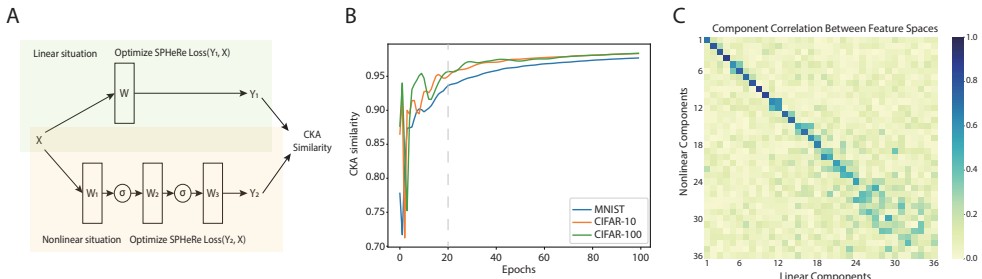

Figure 3: (A) The experiment's network architecture; (B) CKA similarity comparison of the linear and nonlinear branches; (C) SVD component correlation between the linear and nonlinear branches.

## C  Ablation study on Auxiliary Block Architecture

To further evaluate the function of auxiliary blocks, we conduct an ablation study to the auxiliary block. We systematically varied two key components of this design:

1. **Depth:** We changed the number of convolutional layers within $\phi$ (0, 1, or 2 layers).

2. **Projection Dimension:** We changed the output dimension of the final FC projection layer (128, 256, or 512).

The results of this ablation study, measured by test accuracy on CIFAR-10, are summarized in Table 5 and Table 6.

Table 5: Ablation on the number of convolutional layers in $\phi$

| Number of Conv Layers | CIFAR-10 Test Accuracy (%) |
|---|---|
| 0 | 78.65 |
| 1 (Default) | **81.11** |
| 2 | 79.93 |

Table 6: Ablation on the projection dimension of the FC layer in $\phi$

| Projection Dimension | CIFAR-10 Test Accuracy (%) |
|---|---|
| 128 | 80.50 |
| 256 (Default) | **81.11** |
| 512 | 81.18 |

# D Comparison on Transfer Learning Performances

To provide a strong baseline for comparison, we conducted identical experiments using features from a standard Backpropagation (BP) trained model and the current state-of-the-art Hebbian method, SoftHebb [12].

Table 7: Cross-Dataset transfer learning performance comparison. *Non-Transfer* refers to the accuracy achieved by each method when trained and tested on the target dataset (results from Table 1). *Gap* is the difference between *Transfer Learning (Test)* and *Non-Transfer* accuracies.

| Method | Transfer Direction | Transfer Learning (%) | | Non-Transfer (%) |
|---|---|---|---|---|
| | | (Train) | (Test) | |
| BP | Tiny-ImageNet $\rightarrow$ CIFAR-10 | 100.0 | 81.7 | 88.7 |
| | CIFAR-10 $\rightarrow$ Tiny-ImageNet | 99.9 | 39.2 | 45.3 |
| SoftHebb | Tiny-ImageNet $\rightarrow$ CIFAR-10 | 78.0 | 74.84 | 78.86 |
| | CIFAR-10 $\rightarrow$ Tiny-ImageNet | 60.0 | 33.26 | 34.12 |
| SPHeRe (Ours) | Tiny-ImageNet $\rightarrow$ CIFAR-10 | 97.3 | 80.03 | 81.11 |
| | CIFAR-10 $\rightarrow$ Tiny-ImageNet | 99.9 | 37.7 | 40.33 |

# E Comparison with Backpropagation-Based Unsupervised/Self-supervised Learning

To further contextualize the performance of SPHeRe within the broader landscape of unsupervised learning, we compare it against a modern self-supervised learning (SSL) method, SimCLR [**?** ]. This comparison is insightful yet requires careful interpretation due to fundamental differences in learning philosophy and optimal network architecture.

Modern SSL methods like SimCLR leverage strong, explicit pseudo-supervisory signals derived from sophisticated data augmentations to learn invariant and discriminative features. In contrast, SPHeRe operates on a different principle: it uses a simple, unsupervised objective ($\mathcal{L}_{\text{SPHeRe}}$) that aims to preserve the intrinsic structural information of the input without relying on such explicit invariance-inducing signals.

An empirical investigation was conducted on CIFAR-10 using an official SimCLR implementation. Both methods were evaluated under their own optimal hyperparameter settings. The results are summarized in Table 8.

Table 8: Comparison between SPHeRe and SimCLR under different training configurations on CIFAR-10. *Best Settings* refers to the optimal hyperparameters for each respective method. The model structure follows SPHeRe optimal setting.

| Method | Configuration | Train Acc. (%) | Test Acc. (%) |
|---|---|---|---|
| SimCLR | SimCLR Best Settings | 87.83 | 71.94 |
| SPHeRe | SimCLR Best Settings | 75.03 | 70.61 |
| SimCLR | SPHeRe Best Settings | 56.36 | 49.01 |
| SPHeRe | SPHeRe Best Settings | 97.91 | **81.13** |

# F Application to Deeper Architectures

A key question for any unsupervised learning method is its scalability to modern, deep architectures. As noted in the conclusion, our initial observation indicated that the performance gains of purely local SPHeRe learning diminish as network depth increases, due to the accumulation of representation noise across layers. To investigate this quantitatively and explore a practical application, we designed an experiment where a SPHeRe-trained module serves as a biologically-inspired "stem" for a standard deep ResNet.

To evaluate the potential of SPHeRe as an initial feature extractor, we conducted a series of experiments integrating it with a standard ResNet-18 architecture on the CIFAR-10 dataset. We first established a baseline by training a standard ResNet-18 model end-to-end with backpropagation. We then pre-trained the first convolutional layer of the ResNet using the SPHeRe objective. This pre-trained stem was integrated into the ResNet-18 in two ways: 1) by freezing the SPHeRe-trained stem parameters and only training the subsequent ResNet layers, and 2) by using the SPHeRe-trained stem as an initialization and fine-tuning the entire network. The results of these experiments are summarized below.

Table 9: Performance of ResNet-18 on CIFAR-10 with different training strategies for the initial stem layers.

| Model & Training Strategy | Test Accuracy (%) |
|---|---|
| ResNet-18 (Standard Backpropagation) | 79.46 |
| SPHeRe Stem (frozen) + ResNet-18 | 78.78 |
| SPHeRe Stem (fine-tuned) + ResNet-18 | **79.93** |

# G Comparison to related unsupervised methods

This section provides a comparative overview of SPHeRe and several related unsupervised learning and matrix factorization methods. Table 10 summarizes the typical optimization objectives and primary goals for SPHeRe along with typical unsupervised methods such as PCA, Kernel PCA, MDS, SymNMF, and Laplacian eigenmaps. This comparison helps to situate SPHeRe within the broader landscape of dimensionality reduction and structure preservation algorithms, highlighting the similarities and distinctions in their underlying mathematical formulations and objectives.

# H Supplementary Information on the Reconstruction Autoencoder

Fig. 4 shows the framework of the image reconstruction experiments. The encoder shares the same structure as the unsupervised learning setting. The decoder has 3 upsampling Convolution layers ($\{512, 256, 3\}$ channels) with kernel size 3, stride 2, padding 1 and out padding 1. To better visualize the result of the reconstruction, the decoder is trained on the specific image for 100 iterations. We use pixel-wise mean square error to quantify the difference between the reconstructed image and the original image.

To guarantee generalization, we also tested on the whole CIFAR10 testing dataset, for which our method also shows superiority. In this experiment, we trained the decoder on the training dataset for 10 epochs, then tested on the testing dataset. The result is shown in Table. 11.

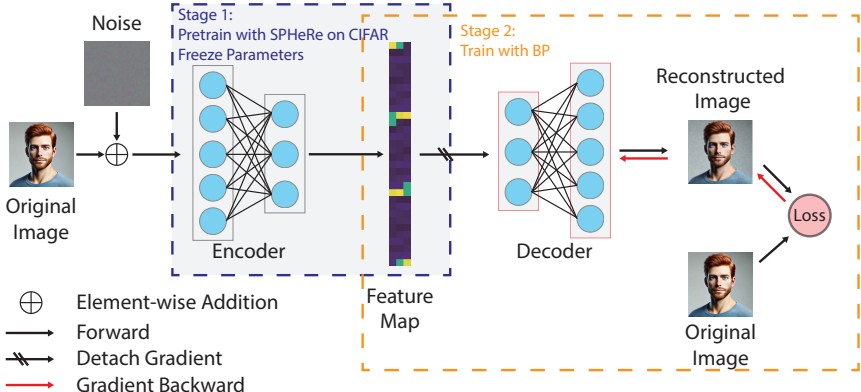

Figure 4: Auto-Encoder framework.

Table 10: Comparison of SPHeRe with Related Unsupervised Learning / Matrix Factorization Methods

| Method | Typical Optimization Objective / Equation | Primary Goal |
|---|---|---|
| **SPHeRe** | $\min_{W,\theta} \|K_Z - K_X\|_F^2$ 
 $Z = \phi(Y,\theta) \quad Y = f(X,W)$ 
 $K_X = XX^\top$ 
 $K_Z = ZZ^\top$ | Preserve input data structure (pairwise relationships defined by $K_X$) in the output representation $Z$. Find optimal low-dimensional projection. |
| PCA | $\max_W \mathrm{Tr}(W^\top X^\top X W)$ 
 s.t. $W^\top W = I$. | Find orthogonal directions (principal components) capturing maximum variance. Dimensionality reduction preserving global covariance structure. |
| Kernel PCA | $\max_V \mathrm{Tr}(V^\top C_\phi V)$ 
 s.t. $V^\top V = I_p$ 
 $C_\phi = \frac{1}{N}\tilde{\Phi}^\top \tilde{\Phi}$ 
 $\tilde{\phi}(x_i) = \phi(x_i) - \frac{1}{N}\sum_{j=1}^{N}\phi(x_j)$ | Find principal components in a non-linear feature space defined by the kernel $k$. Preserve kernel-based structure. |
| MDS | $\min_Y \sum_{i,j} w_{ij}(d_{ij}(Y) - \delta_{ij})^2$ 
 $d_{ij}(Y)$ is distance in embedding $Y$ 
 $\delta_{ij}$ is original distance. | Find a low-dimensional embedding $Y$ that preserves the given pairwise distances or dissimilarities $\delta_{ij}$. |
| SymNMF | $\min_{H\geq 0} \|A - HH^\top\|_F^2$ 
 $A$ is a given symmetric matrix 
 $H$ must be non-negative | Find a non-negative low-rank matrix $H$ such that $HH^\top$ approximates $A$. Often used for clustering. |
| Laplacian Eigenmaps | $\min_Y \mathrm{Tr}(Y^\top L Y)$ 
 s.t. $Y^\top D Y = I$. 
 $(\min_Y \sum_{i,j} w_{ij}\|y_i - y_j\|^2)$ 
 $L = D - W$ | Preserve local neighborhood structure in a low-dimensional embedding. Dimensionality reduction for manifold data. |

Table 11: Reconstruction result on CIFAR10

|  | SoftHebb(100 epoch) | SoftHebb | BP | SPHeRe |
|---|---|---|---|---|
| Loss | 1.81 | 2.39 | 1.68 | **1.62** |

And we provide more results of the image reconstruction tasks in Fig. 5 to support the experimental results and the analysis in the main text.

Note that the reconstruction performance varies across different random initializations. To account for this variability, we repeated the experiment with 10 different random seeds and report the mean reconstruction loss along with the standard error of the mean (SEM) in Fig. 6.

# I   t-SNE Visualization

To qualitatively assess the feature representations learned by the three convolutional layers trained by our method, we visualized the output features for the CIFAR-10 dataset using t-SNE. Fig. 7 presents the t-SNE embeddings for both the training and test set samples, comparing the feature space before training (random weights) and after training. As expected, after training, distinct clusters begin to emerge in the feature space for both training and test sets. This demonstrates that the proposed unsupervised Hebbian learning process successfully organizes the feature representations based on the underlying structure within the CIFAR-10 data, and these learned structures generalize well from the training set to the unseen test set. Consistent with observations made for SoftHebb [12], our SPHeRe approach shows a clear propensity to separate classes characterized by relatively well-defined shapes and edges, such as airplane, automobile, ship and truck. However, it is more challenging to separate

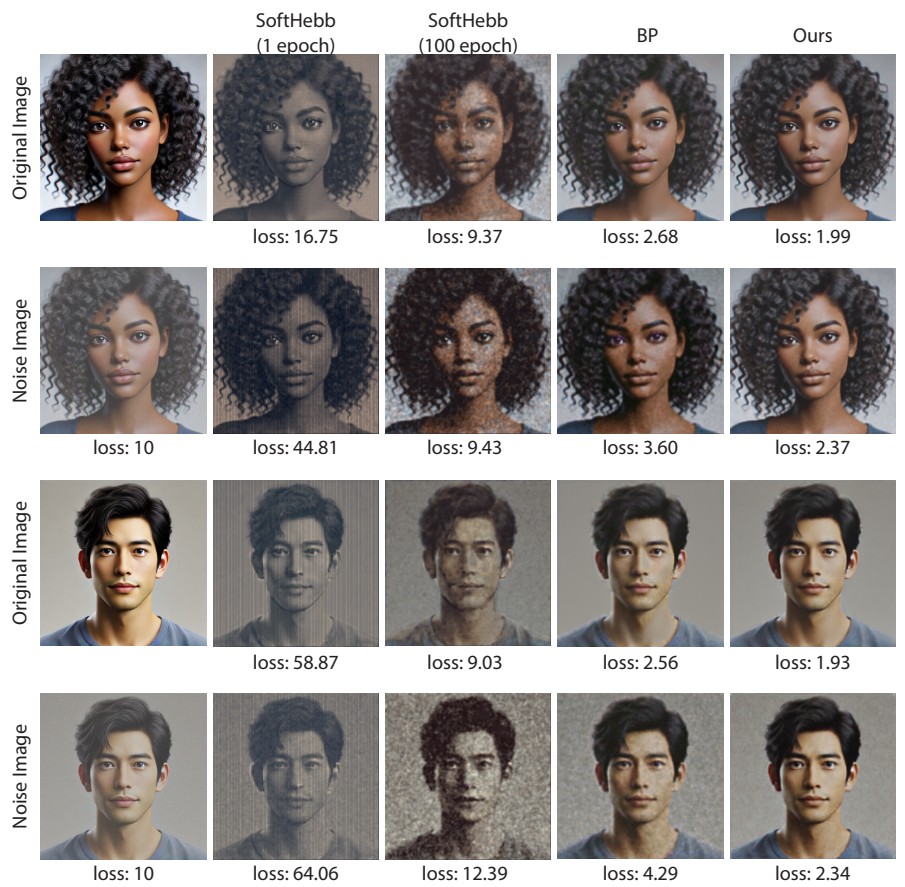

Figure 5: Reconstruction experiment results. The loss values are scaled by a factor of $10^3$.

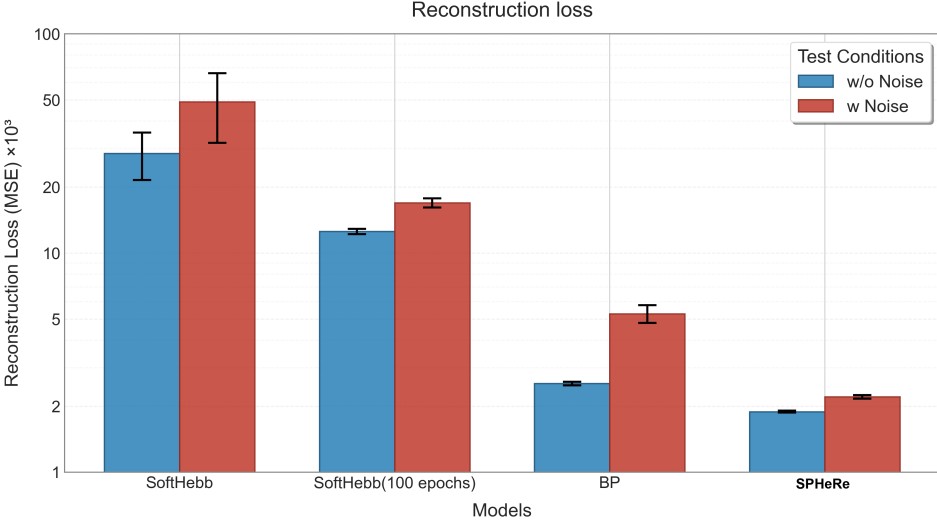

Figure 6: Reconstruction loss across 10 random initializations (mean ± SEM). The y-axis is log-scaled, and the loss values are scaled by a factor of $10^3$.

classes representing animals that often feature fur, feathers, and more variable textures, such as birds, cats, deer, and horses. These classes exhibit considerable overlap in the t-SNE embedding, suggesting that the features captured by SPHeRe, while effective for edge-based discrimination, may be less sensitive to finer textural details or complex shape variations.

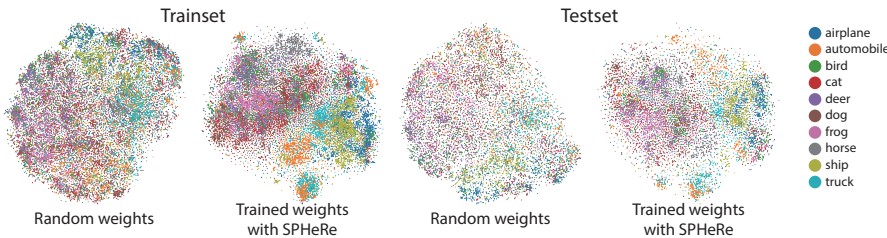

Figure 7: The t-SNE results on CIFAR-10 dataset.

## J  Experiments with Different Activation Functions

To validate the impact of the selection of activation functions on the performance of our model, we conduct experiments considering several common activation functions: ReLU, Leaky-ReLU, tanh, sigmoid, and binary step function. The results are summarized in Table. 12. The performance differences achieved by different activation functions vary significantly, indicating that our approach is still influenced by the choice of activation function. The training set accuracy of ReLU, Leaky-ReLU, and tanh is above 95%, indicating that they are all good choices for the activation function. Among them, Leaky-ReLU performs the best, possibly because it retains the most information before activation while introducing nonlinearity.

Table 12: Comparison for different activation functions

|  | ReLU | Leaky-ReLU | tanh | sigmoid | binary |
|---|---|---|---|---|---|
| Train accuracy | 96.9 | 97.7 | 96.6 | 74.3 | 83.9 |
| Test accuracy | 79.9 | 81.2 | 78.5 | 72.7 | 71.2 |

## K  KNN Experiment

To better monitor the training process, we replace the final classifier with K-Nearest Neighbors (KNN). As shown in Figure 8, the clustering process converges quickly during training. However, after convergence, a slight decrease in accuracy is observed. We assume that this drop is mainly due to overfitting on the training dataset.

## L  Spiking Neural Network Adaption

To further demonstrate the biologically plausible nature of SPHeRe, we demonstrate that our method could work decently on Spiking Neural Networks. The overall network structure is the same with CNN setting, activation function to Leaky-Integrate-and-Fire (LIF) dynamic (with 4 timesteps). We use a surrogate gradient, as spikes are non-differentiable. The surrogate function we choose is the Piecewise-Exponential surrogate function, which can be derived as: $g'(x) = \frac{\alpha}{2} e^{-\alpha|x|}$.

We verify our method and SoftHebb, comparing with random initial weight. The results in Table 13 show that our method can reach decent accuracy, while SoftHebb training has a negative effect on the final result. This may be because softhebb directly implements the Hebbian rule operation on the gradient, without considering the impact of different activation functions on the result distribution, which is also mentioned in the Appendix of the Softhebb paper.

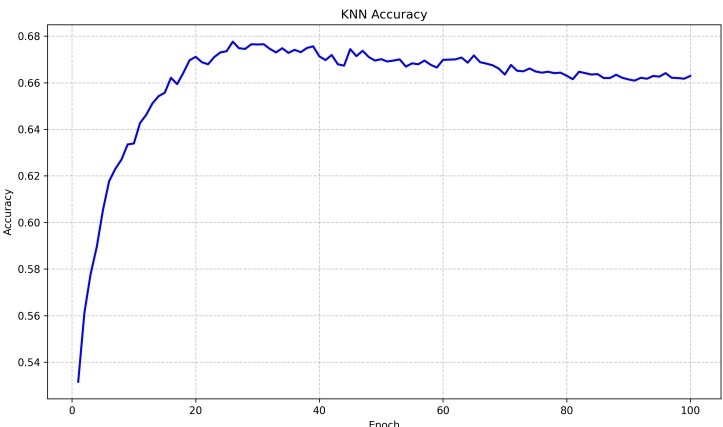

Figure 8: KNN validation accuracy

Table 13: Spiking Neural Network comparison

| Approach | Random weight | SoftHebb | SPHeRe |
|---|---|---|---|
| **Accuracy** | 60.8 | 55.8 | **75.28** |

# M Experiments compute resources

All the experiments are carried out on one Nvidia RTX 3090 Graphic Card with 24GB VRAM. The detailed information is in below:

Table 14: Computational resources used for experiments. Training was conducted on an NVIDIA RTX 3090 GPU (24GB VRAM) with a batch size of 128 and dataloader workers set to 2.

| Dataset | Training Time | GPU VRAM Usage | CPU Workers | Total Epochs |
|---|---|---|---|---|
| CIFAR-10 | ~30 min | ~2.5 GB | 2 | 100 |
| CIFAR-100 | ~30 min | ~2.5 GB | 2 | 100 |
| Tiny-ImageNet | ~3.5 hours | ~8.2 GB | 2 | 100 |

