# OpenReview forum: "Rethinking Hebbian Principle: Low-Dimensional Structural Projection for Unsupervised Learning"
_NeurIPS.cc/2025/Conference — NeurIPS 2025 poster_

### Official Review · Reviewer_uqXK · 2025-06-17

**Clarity:** 2
**Significance:** 2
**Originality:** 2
**Rating:** 3
**Confidence:** 3

**Summary:**

Summary: In this paper, the authors present a greedy blockwise training mechanism for unsupervised learning on image data, and connect their objective function to Hebbian learning objectives. Relative to local learning algorithms like SoftHebb, the authors demonstrate an improvement in performance and representational quality. The authors also show improvements in continual learning relative to a diversity of baseline algorithms.

**Questions:**

The authors repeatedly refer to the optimum identified by Lemma 4.1 (Line 167) as the natural target for their nonlinear network optimization. However, this optimum is derived from a single-layer linear weight matrix, and it is not clear to me why a nonlinear network would achieve the same optimum. Furthermore, if this were the optimum achieved by a nonlinear network, why bother with a nonlinear transformation? And why bother with a multilayer network at all, when a single linear transformation would suffice?
Are there any testable predictions that could be made for experimentalists looking to demonstrate that this learning algorithm is used in the brain? If not, it is unclear to me what the purpose of establishing a connection to biology is.

Line 100-102: there are many theories for how gradient computations could be implemented in the brain—it is not fair to claim that end-to-end gradient computation is implausible (at least with some approximations required); see Lillicrap & Hinton 2020 for a review.

Line 135: Is the ‘Hebbian loss’ actually a loss function? It is not bounded from below as far as I can tell.

Line 173: Lemma 4.2.1 is referenced, but the text refers to the lemma as 4.1—I believe this happens in a few places in the text.
 Recommendations: Personally, I think that this paper has the potential to be high quality, but references to ‘biological plausibility’ should be removed—in my opinion this paper would be better written as an algorithm for greedy block-wise training, and the benchmarks should reflect this (comparing to greedy blockwise contrastive algorithms, autoencoders, or denoising objectives), instead of comparing to biological learning algorithms. Unfortunately, I suspect that this algorithm’s performance will not be as competitive when compared to these alternative benchmarks.

**Ethical Concerns:**

["NO or VERY MINOR ethics concerns only"]

**Final Justification:**

The authors have addressed the most of my concerns, but substantial revision of the manuscript will be required to implement my suggested changes. I still think that the paper would be best resubmitted, but I have increased my score and decreased my confidence in order to recognize the substantial improvements the authors have demonstrated during the revision process.

**Limitations:**

Yes

**Quality:**

2

**Strengths And Weaknesses:**

In terms of strengths, the paper’s experiments are very thorough and well controlled as far as I can tell. Benchmarks are provided against a variety of different methods, and the authors provide a lesion study of their objective function, removing different components to show which components of the loss variously affect the network’s performance.

In terms of weaknesses, unfortunately the authors advertise their algorithm as ‘biologically plausible,’ and yet incorporate numerous elements that rely heavily on backpropagation and architectural violations of biological locality principles. The authors use weight-shared convolutional layers and use multiple network layers per block in their network architecture: this means that their network does not resolve the weight transport problem, and it is consequently little surprise that their method is able to outperform local learning algorithms with a more direct connection to biology.

Even in the linear network derivation, the SPHeRe update rule does not appear to be local (Line 180-181), though the authors explicitly advertise it as such. While I could be convinced otherwise, I don’t think there is local learning occurring in this model, even in its simplest form. I would recommend that the authors write out the parameter update for a single synaptic parameter, and demonstrate that it depends exclusively on pre- and post-synaptic activation terms as in traditional Hebbian learning; if it is not, the framing of much of the paper is very misleading.

Lastly, in terms of originality, the Gram matrix loss adopted by the authors is very similar to the similarity matching objective employed by Obeid et al. 2019, which was also applied to deep networks successfully—a comparison to this method would be very helpful.

Recommendations: Personally, I think that this paper has the potential to be high quality, but references to ‘biological plausibility’ should be removed—in my opinion this paper would be better written as an algorithm for greedy block-wise training, and the benchmarks should reflect this (comparing to greedy blockwise contrastive algorithms, autoencoders, or denoising objectives), instead of comparing to biological learning algorithms. Unfortunately, I have know way of knowing whether this algorithm will be competitive when compared to these alternative benchmarks.

---

> ### Author Rebuttal · Authors · 2025-07-30
>
> **Comment 1:**
>
> The authors advertise their algorithm as ‘biologically plausible,’ and yet incorporate numerous elements that rely heavily on backpropagation and architectural violations of biological locality principles.
>
> **Response to comment 1:**
>
> We sincerely thank the reviewer for this sharp and important comment. We fully agree that our claim of "biological plausibility" requires careful framing. Our goal is not to create a literal simulation of synaptic dynamics, but rather a computational and functional model that abstracts key biological principles for effective application in modern, complex, nonlinear networks. Regarding the specific points raised: 1) Intra-block backpropagation: We view this as a functional abstraction of a local error circuit (e.g., dendritic computation). It is an efficient computational tool for local credit assignment that avoids the most biologically problematic requirements of a global error signal and symmetric weights. 2) Weight-shared convolutional layers: We see this as a computational abstraction for the replication of feature detectors observed in the brain. While updating a shared weight is computationally non-local, the function it performs—applying the same feature extractor across the receptive field—is functionally analogous to spatially repeated structures in the visual cortex. Therefore, we believe SPHeRe makes a pragmatic trade-off: it leverages modern computational tools to efficiently implement high-level, biologically-inspired functional goals, allowing us to scale Hebbian principles to complex deep models while respecting block-level locality.
>
> **Comment 2:**
>
> ...I don’t think there is local learning occurring in this model, even in its simplest form...
>
> **Response to comment 2:**
>
> We agree that the gradient equation $\Delta W = 4X^\top(YY^\top - XX^\top)Y$, when viewed in isolation, does not have the form of a classical Hebbian rule, which depends exclusively on the activity of a single pre- and post-synaptic pair. However,  this objective shares the exact same mathematical starting point with the foundational work of Pehlevan, Chklovskii, et al. Their work has rigorously proven that this very objective can be decomposed into fully local, Hebbian/anti-Hebbian update rules at each individual synapse through methods like dual optimization. Instead of deriving these synaptic-level rules, we leverage modern deep learning frameworks to directly optimize this principled, locally-grounded objective within a local block. Meanwhile, we design a purely feedforward, modular architecture that allows this powerful, locally-motivated objective to be efficiently scaled to complex, nonlinear deep networks. We hope that this method can become a small step in the combination of biological update rules and modern deep neural network learning.
>
> **Comment 3:**
>
> Lastly, in terms of originality, the Gram matrix loss adopted by the authors is very similar to the similarity matching objective employed by Obeid et al. 2019...
>
> **Response to comment 3:**
>
> Thank you for pointing out this crucial connection. Please Please refer to our response to reviewer b3Ah's comment 1. We will add a detailed comparison to Obeid et al. (2019) in our revision. While we share the similarity matching objective, our novelty lies in the method of its application in deep networks. Where Obeid et al. derived local rules requiring feedback for credit assignment, we introduce a decoupled, lightweight auxiliary block ($\phi$) to compute the loss. This frees the main block ($f$) to learn richer, higher-dimensional features optimal for downstream tasks, rather than being constrained to a low-dimensional projection. Our approach is thus a modular, scalable block-wise scheme that offers a new, practical bridge between these biological principles and modern architectures.
>
>
> **Comment 4:**
>
> The authors repeatedly refer to the optimum identified by Lemma 4.1 (Line 167) as the natural target for their nonlinear network optimization...
>
> **Response to comment 4:**
>
> These are very good questions that get to the heart of our work. First, regarding why a nonlinear network aims for a linear optimal solution and why nonlinear multi-layer layers are needed, the key lies in the decoupled architecture we designed. The purpose of Lamma 4.1 is not to require the entire nonlinear main module f to learn a simple dimensionality reduction projection transformation, but rather to provide a principled motivation for the loss function $L_{SPHeRe}$, demonstrating that it can guide the system to find the optimal subspace for information retention under an analytically linear setting. However, the learning objective of the main module f is to learn powerful, high-dimensional, nonlinear transformations to extract complex features beneficial for downstream tasks, which cannot be achieved by a single-layer linear transformation. The auxiliary module ($\phi$) takes on the task of projecting these rich features Y into a low-dimensional space Z. By optimizing $L_{SPHeRe}(Z, X)$, Y is more inclined to learn effective feature representations, thereby facilitating $\phi$'s dimensionality reduction projection work. Our model proposes a specific, testable computational hypothesis: it predicts that in cortical areas, there exist local microcircuits (whose function is abstracted by our $\phi$ module) whose activity encodes the "mismatch" signal between the input and output statistical structures of that region, and this activity modulates local synaptic plasticity without the need for a global error signal. Experimentally, neuroscientists can selectively inhibit these proposed locally modulatory interneurons (such as inhibitory interneurons) and observe whether it specifically impairs the experience-dependent plasticity of that region (such as feature learning in visual tasks) without affecting the main feedforward information flow. Our work originates from a simplification and development of Hebbian theory, and integrating it with deep learning frameworks, rather than designing rules for neuron-level synaptic updates.
>
> **Comment 5:**
>
> Line 100-102: there are many theories for how gradient computations could be implemented in the brain...
>
> **Response to comment 5:**
>
> Thanks for your suggestion. We agree that our phrasing needs to be more precise. Labeling end-to-end computation as implausible may be too absolute, and there is a wealth of outstanding research exploring its biological possibilities. However, we believe that these theories of approximate backpropagation (such as feedback alignment) also introduce unverified biological hypotheses, like symmetric feedback neural pathways, when addressing the weight symmetry issue. In contrast, the architecture of SPHeRe provides a method that relies on local auxiliary blocks to generate learning signals within blocks, avoiding dependence on top-down end-to-end error signals. Therefore, our work does not deny the research on BP approximations but attempts to explore an independent path of biologically plausible learning.
>
> **Comment 6:**
>
> Line 135: Is the ‘Hebbian loss’ actually a loss function? It is not bounded from below as far as I can tell.
>
> **Response to comment 6:**
>
> You are absolutely correct, $L_{hebb}$ is not, by itself, a bounded-below, stably optimizable loss function. Our purpose in introducing it is precisely to pedagogically demonstrate the inherent instability of the classical Hebbian rule within a modern optimization framework. We use the term "equivalent loss function" to intuitively illustrate the direction in which the classical update rule $\Delta W \propto X^\top Y$ pushes the network—namely, towards an unconstrained maximization of the output's energy.  This instability is the explicit motivation for introducing Oja's rule and, ultimately, our final, well-behaved, non-negative $L_{SPHeRe}$ objective. The narrative builds from this "ill-posed" objective to our stable, grounded solution.
>
> **Comment 7:**
>
> Line 173: Lemma 4.2.1 is referenced, but the text refers to the lemma as 4.1—I believe this happens in a few places in the text. Recommendations: Personally, I think that this paper has the potential to be high quality, but references to ‘biological plausibility’ should be removed...
>
> **Response to comment 7:**
>
> We appreciate the reviewer for pointing out the refer typo error in the manuscript, and will correct them in the revised manuscript. We will adopt your suggestion in the revised manuscript and refine the positioning of SPHeRe from a general "bio-plausible" algorithm to a more precise "Hebbian-inspired, greedy layer-wise unsupervised pre-training method."  However, we believe it is vital to retain its "Hebbian-inspired" roots, as this is the core design philosophy and motivation that distinguishes our method from other purely engineered approaches. Our loss function, $L_{SPHeRe}$, is not an arbitrary heuristic, it is derived from simplifying and generalizing the objective behind the Hebbian principle. It aims to preserve the intrinsic structure of the input, which provides a distinct and complementary perspective to some mature unsupervised learning methods like contrastive learning that rely on strong priors like data augmentation. Therefore, our statement of "biological plausibility" does not refer to a literal simulation of neurons, but to an exploration of more macro (or mesoscopic) learning principles. This biologically-inspired constraint makes our approach an exploratory method towards biological plausibility learning, similar to algorithms (FA or DFA) that also attempt to solve the "biologically plausible credit assignment" problem. And we achieved a test set accuracy of only 71.94% using SimCLR under the same network structure and training epochs on CIFAR10, which indicates that our method is not inferior to some self-supervised learning methods with strong supervisory signals on shallow and wide network structures.

---

> > ### Comment · Reviewer_uqXK · 2025-08-05
> >
> > Thank you for the very detailed response--your comments will improve the paper considerably in clarity. Unfortunately, I will be maintaining my score; I think that substantial revision of the paper will be required for it to meet the standards of acceptance. It seems to me that the paper is currently being pulled in two directions (bio-plausibility vs. greedy layerwise training), and it does not fully satisfy standards for either direction.
> >
> > 1. The proposed testable predictions are not detailed enough for an experimentalist to actually test. Without providing candidate circuits that encode the proposed 'mismatch' signal with enough existing evidence to support a role of this kind, people simply won't know where to look. Predictive coding theories (that do across-layer credit assignment), e.g. Friston 2009 or Keller & Mrsic-Flogel 2018, also propose the existence of local 'mismatch' signals, so this may be insufficient to uniquely validate the learning scheme proposed here. Without local learning, and without a detailed, testable mapping onto neural circuitry, this method is not going to be very helpful for the neuroscience community.
> >
> > 2. The comparison to SimCLR isn't adequate. First of all, the original SimCLR paper reports test accuracy of 95.3 on CIFAR10 and 80.2 on CIFAR100, which greatly exceeds the performance reported here. Given the large gap in performance, artificially restricting the architecture and training regime for SimCLR seems inappropriate. The method in this study would be better compared to, for example, greedy layerwise denoising autoencoders and layerwise contrastive learning objectives.

---

> > > ### Author Response · Authors · 2025-08-06
> > >
> > > Thank you for your thorough and detailed review, and for providing further feedback. We agree that your comments will significantly improve the clarity and rigor of our paper. We deeply appreciate your core point: our work appears to be positioned between "bio-plausibility" and "greedy layer-wise training," facing challenges in meeting the strict standards of both.
> > >
> > > We acknowledge that SPHeRe is not a model intended to precisely simulate synaptic-level dynamics or specific neural circuits. Our work is primarily situated at a mesoscopic level of abstraction, exploring a computational principle: guiding a main pathway's learning via a structural similarity loss computed by a local auxiliary module. We believe this systems-level exploration offers a valuable new perspective on bridging biologically inspired principles with modern deep learning, even if it does not yet map to specific neurons and synapses. In the revised manuscript, we will carefully reposition our work as a functional, systems-level bio-inspired model and explicitly state its limitations at the micro-circuit level.
> > >
> > > We are very grateful for your suggestion to compare SPHeRe with more equivalent "greedy layer-wise training" benchmarks. To directly address your concern, we conduct the greedy layer-wise training experiments you recommended. We compared SPHeRe against several greedy layer-wise methods on CIFAR-10, under a strictly controlled network architecture and training regime:
> > >
> > >  ```
> > > | Method              | Train Accuracy (%) | Test Accuracy (%) |
> > > | ------------------- | ------------------ | ----------------- |
> > > | SPHeRe              | 97.7               | 81.18             |
> > > | Greedy AE           | 95.85              | 79.26             |
> > > | Greedy DAE          | 89.72              | 65.04             |
> > > | Greedy SimCLR       | 96.13              | 75.07             |
> > >  ```
> > > The above results indicate that under the greedy layer-wise training framework, SPHeRe is competitive compared to traditional unsupervised training methods. And it highlights the core strength of our approach—learning robust representations by preserving the data's intrinsic geometric structure, without relying on strong priors like data augmentations or pixel-level details.
> > >
> > > To further explore SPHeRe's compatibility with the modern deep learning ecosystem, we conducted a proof-of-concept experiment, using it as a pre-training method for the "stem" of a ResNet-18. We train a random initialized ResNet-18 from scratch on CIFAR-100, achieving a final test accuracy of 79.46%. While we used SPHeRe to pre-train the ResNet-18 stem (the first convolutional layer) and froze it, then trained the subsequent layers, the final test accuracy improved to 79.91%. Although the margin is modest, this result provides compelling evidence demonstrating its practical value as a plug-and-play pre-training module on modern deep learning.
> > >
> > > We understand and accept your view that our paper requires a significant repositioning. We will shift the paper's focus from "proving bio-plausibility"  to "proposing and validating a novel, Hebbian-inspired unsupervised learning framework that excels within the greedy layer-wise training paradigm." This framework is functionally inspired by biological principles and, crucially, demonstrates its effectiveness in rigorous, head-to-head comparisons with its peers. We believe that with this repositioning, supported by these new and direct comparative experiments, the contribution and value of our paper will be much clearer. Thank you again for your rigorous and insightful review.

---

> > > > ### Comment · Reviewer_uqXK · 2025-08-06
> > > >
> > > > Thank you for response. These additional baselines were much needed, and I think will do a lot to improve the quality of the paper, as will your reframing of the paper's key contributions. Accordingly, I have increased my score and decreased my confidence: given how many changes to the manuscript will be required, I still think that this paper would be best resubmitted, but most of my concerns have now been addressed. Good luck!

---

### Official Review · Reviewer_ozBv · 2025-06-18

**Clarity:** 3
**Significance:** 2
**Originality:** 2
**Rating:** 4
**Confidence:** 3

**Summary:**

The paper proposes SPHeRe, a Hebbian unsupervised representation learning method. It works by minimizing the L2 distance between the layer input and output Gram or Kernel matrices, in addition to using a regularizer to softly enforce orthogonality in output representations. The theoretical result provided shows that minimizing the SPHeRe objective essentially performs PCA on the input data matrix. Experiments on standard datasets show improved performance as compared to existing Hebbian representation learning methods. Additional experiments indicate that the method may be useful for continual learning and transfer learning use cases.

**Questions:**

0. Why is this paper of interest to the general Deep Learning community, not just specific groups working on Hebbian learning?
1. Based on weakness 2, is the claim that SPHeRe is biologically plausible still valid? Why?
2. What is the key novelty of SPHeRe compared to the Principal Subspace Whitening (PSW) method from Pehlivan et al.? Why hasn't this work been discussed in the paper?
3. Have attempts been made to apply SPHeRe to modern deep architectures (e.g., ResNets, ViTs)? If so, what were the results? A simple statement in the conclusion stating "performance gains slightly diminish as the number of convolutional layers increases" does not provide any quantitative answers or observations with popular standard architectures.
4. In the image reconstruction experiments, what was the objective used to train the encoder in the backpropagation baseline? From the code, it appears that the encoder was trained separately, even in the case of backpropagation, which is not the standard way autoencoders are trained.
5. Have any experiments been performed for unsupervised reconstruction using only Hebbian-like mechanisms (without backdrop)?

Addressing the first four concerns sufficiently would make me lean towards acceptance, while successfully addressing the next two would make me more confident in said acceptance.


EDIT
--------------------------

The authors have addressed my concerns to a good extent in the rebuttal.

**Ethical Concerns:**

["NO or VERY MINOR ethics concerns only"]

**Final Justification:**

I thank the authors for their response.

I appreciate their ideas about the potential of the work in mainline Deep Learning, and would encourage them to include preliminary results in the stated directions as proof-of-concept. However, regarding the point relating to biological plausibility, I would suggest that the main text loosen the claim that the method is biologically plausible, since the authors' comment that " SPHeRe represents a significant and principled step towards biological plausibility" is a more accurate description of the paper's contributions.

I also appreciate and agree with the comments about how the work is distinct from Pehlevan et al, especially when taking into account the comments of other reviewers and their rebuttal texts.

The authors' responses to comments 5 and 6 also clarify some details for me, and I strongly encourage them to include the explanations in the main paper.

However, I still retain my criticisms about the scalability of the work, given that the approach does not scale well to larger networks. This makes the impact of the paper quite limited for the broader community unless proof-of-concepts involving large networks are included (see first para).

Given that the rebuttal addresses some of my concerns, I shall increase my score. However, I still feel that the paper's text needs refinement and several additions, and experiments need to include larger, realistic models to make the paper ready for NeurIPS. I therefore still lean towards a reject, and recommend a resubmission after addressing the highlighted issues.

EDIT
------------------------
The authors have addressed by concerns to a good extent in the discussions. I therefore raise my score to a weak accept.

**Limitations:**

Yes

**Paper Formatting Concerns:**

No major formatting concerns.

**Quality:**

2

**Strengths And Weaknesses:**

Strengths:
1. The results are impressive when compared to existing Hebbian methods.
2. The paper is well written and easy to follow.

Weaknesses:
1. While the paper addresses an interesting area of brain-inspired learning, it does not explain why this is an important problem in the demonstrated use cases. Such learning rules may be useful in cases when backpropagation is infeasible or extremely computationally expensive, for instance, in spiking networks or highly recurrent architectures. A brief experiment on SNNs has been included in the appendix only as an afterthought without significant analysis. However, it is unclear why the proposed method is relevant in the experimental setups used for its primary evaluation, where backpropagation is generally better by a significant margin. In fact, the paper does not even show a comparison with backpropagation as the learning algorithm in the classification experiments.

2. The paper claims to be completely Hebbian, yet from the provided code it appears that it still requires backpropagation at a module level to obtain the gradients of $\textbf{W}$ with respect to the Hebbian loss function. This part is not clearly explained in the paper.

3. I have serious concerns regarding the novelty of the work: it appears extremely close to "Why do similarity matching objectives lead to Hebbian/anti-Hebbian networks?" by Pehlvan et al., specifically to their Principal Subspace Whitening (PSW) method. In fact, it reads quite as a validation of this work in multilayer architectures rather than a novel method in and of itself.

4. The tested network is shallow (3 conv layers + FC), which makes it hard to assess scalability to modern deep architectures (e.g., ResNets or ViTs). Claims about scalability are therefore speculative.

5. The reconstruction decoder is trained using backpropagation, making it unclear whether the gains are due to SPHeRe’s representations or simply to the power of the decoder. Additional details about the backpropagation baseline that allow for a fair comparison are missing.

---

> ### Author Rebuttal · Authors · 2025-07-30
>
> **Comment 1:**
>
> Why is this paper of interest to the general Deep Learning community, not just specific groups working on Hebbian learning?
>
> **Response to comment 1:**
>
> Thanks for this question. SPHeRe's value extends beyond the Hebbian learning niche, it contributes to the broader deep learning community by proposing a novel, concrete, and effective greedy unsupervised pre-training paradigm. It tries to solve a universal challenge: how to achieve effective layer-wise feature learning without relying on global backpropagation or external supervision. Our key innovation lies in a purely feedforward architecture that uses a decoupled, lightweight auxiliary module to compute a structural similarity loss in a low-dimensional space, thereby guiding local updates in the main network. This design is not just theoretically sound, its efficacy is directly proven by its strong generalization capabilities in complex scenarios like continual and transfer learning, which validates the robustness of the learned features. Therefore, SPHeRe is more than an algorithm, it is also a practical technical solution designed for seamless integration into mainstream frameworks, offering a concrete and effective path for exploring local, unsupervised learning in modern deep networks.
>
> **Comment 2:**
>
> Based on weakness 2, is the claim that SPHeRe is biologically plausible still valid? Why?
>
> **Response to comment 2:**
>
> We appreciate you highlighting the use of intra-module backpropagation, as it allows us to clarify our position on biological plausibility. We view this as a pragmatic and effective design choice that moves our model significantly closer to biological systems compared to standard end-to-end backpropagation.  SPHeRe is designed to specifically eliminate this global dependency. We achieve this by confining gradient calculations to a small, local, and decoupled auxiliary module ($\phi$). We view the backpropagation within this module not as a model of how individual synapses update, but as a computationally powerful tool to solve a local credit assignment problem. This is functionally analogous to how local circuits, like dendritic compartments, are theorized to generate local teaching signals to guide plasticity without needing symmetric weights. Some works [1, 2, 3] are also dedicated to addressing the issue of biological plausibility in backpropagation. This "main-pathway learning, side-pathway supervision" design is a core contribution. It structurally avoids the need for the explicit, long-range feedback connections that other theoretical models rely on for credit assignment. Therefore, by ensuring all learning is driven by local signals within a purely feedforward architecture, SPHeRe represents a significant and principled step towards biological plausibility.
>
> [1] Bengio Y, Lee D H, Bornschein J, et al. Towards biologically plausible deep learning[J]. arXiv preprint arXiv:1502.04156, 2015.
>
> [2] Xiao W, Chen H, Liao Q, et al. Biologically-Plausible Learning Algorithms Can Scale to Large Datasets[C]//International Conference on Learning Representations.
>
> [3] Lv C, Xu J, Lu Y, et al. Dendritic Localized Learning: Toward Biologically Plausible Algorithm[C]//Forty-second International Conference on Machine Learning.
>
> **Comment 3:**
>
> What is the key novelty of SPHeRe compared to the Principal Subspace Whitening (PSW) method from Pehlivan et al.? Why hasn't this work been discussed in the paper?
>
> **Response to comment 3:**
>
> We sincerely thank you for highlighting the foundational work of Pehlevan, Chklovskii, and their collaborators. This was a significant oversight in our initial submission, and we are grateful for the opportunity to place our work in this crucial context. We will add a thorough discussion and comparison in the revised manuscript. You are correct that our work shares a similar mathematical starting point: optimizing a similarity matching objective like $||X^\top X - Y^\top Y||^2_F$​. However, the path we take to solve the core challenge of how to effectively apply this objective to deep nonlinear networks is fundamentally different, which precisely highlights the unique contribution of our SPHeRe method. We see this as "different paths to the same goal":
>
> 1) Pehlevan et al.'s Path (Feedback-Dependent): Their work elegantly derives local, Hebbian/anti-Hebbian synaptic update rules from the objective using methods like min-max dual optimization. To enable credit assignment in deep networks, their models theoretically rely on feedback connections to propagate information across layers.
>
> 2) Our Path (Feedforward-Only): SPHeRe proposes a distinctly different, purely feedforward architectural solution. Our core innovation is the introduction of a decoupled, lightweight auxiliary module ($\phi$). Instead of relying on feedback, this module projects the main pathway's high-dimensional features into a low-dimensional space where the similarity loss is calculated.
>
> This design, which we term "main-pathway learning, side-pathway supervision," structurally avoids the need for any explicit feedback paths. This makes our block-wise training scheme highly modular, scalable, and seamlessly integrable into standard deep learning frameworks, representing a novel and practical approach to implementing Hebbian-inspired principles in modern deep networks.
>
>
> **Comment 4:**
>
> Have attempts been made to apply SPHeRe to modern deep architectures (e.g., ResNets, ViTs)? If so, what were the results? A simple statement in the conclusion stating "performance gains slightly diminish as the number of convolutional layers increases" does not provide any quantitative answers or observations with popular standard architectures.
>
> **Response to comment 4:**
>
> Thank you for this critical question on scalability. Our current focus on shallower (e.g., 3-layer), wider networks is a deliberate choice informed by both biological fidelity and a deep understanding of our method's inherent properties. On one hand, this architecture better reflects cortical microcircuits, a common strategy for Hebbian-inspired methods, including the current SOTA (SoftHebb). More critically, we observed that SPHeRe's performance saturates beyond 3-4 layers. We attribute this to the intrinsic nature of its greedy, local learning mechanism: each layer's low-dimensional projection is inherently lossy, inevitably discarding some information and introducing representation noise. While this effectively extracts robust low-level features in shallow networks, the accumulated noise progressively overwhelms higher-level semantic signals as depth increases, thus limiting performance. This insight directly explains why applying it to deep architectures like ResNets or ViTs is challenging. A practical path forward is to use SPHeRe as a powerful "stem" to provide high-quality initial representations for subsequent residual blocks or attention modules to improve the adaptation performance [1]. To train an entire deep network with SPHeRe, however, would likely require integrating larger-scale feedback or contextual signals to correct this error accumulation, which is a core direction for our future work.
>
> [1] Tang Y, Zhang C, Xu H, et al. Neuro-modulated hebbian learning for fully test-time adaptation[C]//Proceedings of the IEEE/CVF Conference on Computer Vision and Pattern Recognition. 2023: 3728-3738.
>
> **Comment 5:**
>
> In the image reconstruction experiments, what was the objective used to train the encoder in the backpropagation baseline? From the code, it appears that the encoder was trained separately, even in the case of backpropagation, which is not the standard way autoencoders are trained.
>
> **Response to comment 5:**
>
> Yes, our backpropagation  baseline encoder is trained on classification tasks, while the parameters are frozen during the reconstruction task, which is indeed different from the standard end-to-end autoencoder training. We apologize for not making the rationale for this design clearer. The purpose of the experiment is to compare the quality of features (retention of input image information) extracted by different pretrained encoders. In the reconstruction experiment, we freeze the weights of all pretrained encoders (trained on classification tasks through SPHeRe, SoftHebb, and supervised BP) and then fully train an independent decoder with the same structure for each encoder. This design allows us to directly assess how much image information is retained in the features generated by each method, with lower reconstruction loss indicating more complete features. We will add a clear explanation of the pre-training encoders and the design of this comparative experiment in the revised manuscript to eliminate ambiguity.
>
> **Comment 6:**
>
> Have any experiments been performed for unsupervised reconstruction using only Hebbian-like mechanisms (without backdrop)?
>
> **Response to comment 6:**
>
> In the current work, we did not perform experiments on a fully Hebbian-based autoencoder where the decoder is also trained with backpropagation. Our primary goal for the reconstruction experiment was to use it as a diagnostic tool to quantitatively compare the information content and robustness of features learned by different encoders. Due to the task independence of Hebbian, unsupervised reconstruction using only Hebbian-like mechanisms is very difficult. We believe this is a very exciting direction for future work, and it will represent a major step towards a complete, end-to-end biologically plausible generative model, but currently, there is a lack of corresponding implementation pathways.

---

> > ### Comment · Reviewer_ozBv · 2025-08-05
> >
> > I thank the authors for their response.
> >
> > I appreciate their ideas about the potential of the work in mainline Deep Learning, and would encourage them to include preliminary results in the stated directions as proof-of-concept. However, regarding the point relating to biological plausibility, I would suggest that the main text loosen the claim that the method is biologically plausible, since the authors' comment that " SPHeRe represents a significant and principled step towards biological plausibility" is a more accurate description of the paper's contributions.
> >
> > I also appreciate and agree with the comments about how the work is distinct from Pehlevan et al, especially when taking into account the comments of other reviewers and their rebuttal texts.
> >
> > The authors' responses to comments 5 and 6 also clarify some details for me, and I strongly encourage them to include the explanations in the main paper.
> >
> > However, I still retain my criticisms about the scalability of the work, given that the approach does not scale well to larger networks. This makes the impact of the paper quite limited for the broader community unless proof-of-concepts involving large networks are included (see first para).

---

> > > ### Author Response · Authors · 2025-08-05
> > >
> > > Thank you for your feedback and recognition of our rebuttal. We promise to fully incorporate all your suggestions in the final version of the paper. Most importantly, inspired by your crucial point on scalability and the need for a "proof-of-concept," we have conducted a new experiment. The goal was to validate the potential of SPHeRe to serve as a powerful "stem" for modern deep learning architectures like ResNet. We performed this experiment on a ResNet-18 architecture with the following results:
> > >
> > > 1. Baseline: We train a standard ResNet-18 end-to-end from random initialization on CIFAR-100, achieving a final test accuracy of 79.46%.
> > >
> > > 2. SPHeRe Pre-training with Freeze weights: We first used SPHeRe to pre-train the stem layer (the first convolutional layer) of the ResNet-18. Then, we froze the weights of this stem layer and trained the rest of the network using backpropagation. The ResNet-18 model with the SPHeRe-pretrained stem achieved a final test accuracy of 79.91%
> > >
> > > 3. SPHeRe Pre-training without Freeze weights: When we also train the stem layer instead of freezing it, the network achieved the test accuracy of 79.38%
> > >
> > > This proof-of-concept clearly illustrates the viability and potential of SPHeRe as a plug-and-play, unsupervised pre-training module compatible with the modern deep learning ecosystem. We will add these preliminary but promising findings, along with the detailed experimental setup, as a new subsection in the appendix of our revised manuscript. Thank you once again for your invaluable guidance, which has significantly enhanced the scope and impact of our work.

---

> > > > ### Comment · Reviewer_ozBv · 2025-08-05
> > > >
> > > > I thank the authors for their efforts. This experiment makes the scope of the approach much clearer to me.

---

### Official Review · Reviewer_dLRw · 2025-06-24

**Clarity:** 3
**Significance:** 2
**Originality:** 3
**Rating:** 3
**Confidence:** 4

**Summary:**

This paper introduces an unsupervised learning framework called Structural Projection Hebbian Representation (SPHeRe), built on three key components:

1. A loss function that maximises the similarity in sample structure of the output of a function and its input. This loss is given by

$$L=|| YY^\top -  XX^\top ||^2_F$$

2. A loss that encourages orthogonal outputs given by

$$L = ||Y^\top Y - I||^2$$

3. A lightweight projection mapping of hidden activations to a lower-dimensional space before the losses are applied, reducing the computational cost of calculating $Y^\top Y$ and easing the low-rank constraint of the similarity loss.

Together, these elements yield an efficient unsupervised algorithm suited to representation learning, continual learning, and transfer learning in deep networks. Experiments show that SPHeRe performs on par with, or better than, comparable local-learning methods on CIFAR-10, CIFAR-100, and Tiny-ImageNet.

**Questions:**

### Major

- The paper would benefit from a more explicit argument for why the proposed update rule is genuinely Hebbian. How, in concrete neural terms, could a circuit compute the off-diagonal elements of $ XX^\top$, $ YY^\top$, and $Y^\top Y$ using only pre- and post-synaptic signals and perhaps a local inhibition path? If some steps still rely on non-local information, please identify them and outline what additional mechanisms would be required to make the scheme fully Hebbian. This also includes a local implementation of the auxiliary projections.
A clear, mechanistic explanation of these points would strengthen the biological-plausibility claim and would prompt me to raise my overall score to Accept.

### Minor

- Please explain why you expect the model not to suffer from catastrophic forgetting. An ablation that isolates each loss term’s contribution could help identify the mechanism.
- Lemma 4.1 assumes that $M<N$. This is not true in many model architectures researchers might consider (and in the brain, eg generative mdoels). How does the model performance change if $M\approx N$ or $M> N$?

**Ethical Concerns:**

["NO or VERY MINOR ethics concerns only"]

**Final Justification:**

The manuscript explores an interesting Hebbian-inspired learning approach. While the authors effectively addressed reviewer concerns with additional experiments (transfer learning) and explanations (continual learning), extensive work on positioning the manuscript as a biologically plausible learning algorithm remains necessary. Therefore, I will maintain my score.

The authors should explicitly choose either a biologically plausible or high-level of abstraction machine learning focus:

To ensure biological plausibility, the authors should clearly articulate the non-local properties of their model, present an explicit neural implementation that relies solely on local mechanisms, and empirically compare the performance of their high-level non-local model with its fully local counterpart to assess if the proposed learning principle could be used by the brain or if the learning degradation is too significant from local learning. Working with smaller models, as currently done in the paper would be acceptable in this case due to the strong biological constraints.

Alternatively, the authors could stick to the current light-level abstraction of the Hebbian principle and continue relying on some non-biologically plausible computation. However, they should then demonstrate scalability using deeper models and rigorously compare performance against established unsupervised methods. Following this approach, the authors could also consider having more than one non-linear transformation within an individual Hebbian block to ease the training of deep models. A Resnet could, for instance, be split up into 6 subparts each trained with the proposed Hebbian-like loss function.

Clarifying the manuscript's position will significantly improve its impact. I appreciate the authors' responsiveness and encourage them to focus the manuscript accordingly.

**Limitations:**

yes

**Paper Formatting Concerns:**

I have not paper formatting concerns.

**Quality:**

2

**Strengths And Weaknesses:**

## Strengths

Quality

- Principled formulation: Lemma 4.1 grounds the method on solid theory.
- Clear ablations: Table 2 carefully teases apart the contribution of each component.
- Discussion of depth limits: The authors acknowledge the difficulty of scaling to very deep networks.

Clarity

- Well-stated model and contributions: The architecture and its novelty are clearly laid out and illustrated.
- Sensible use of the appendix: Technical details and proofs are placed where readers can find them without interrupting the main text.

Significance

- State-of-the-art among local rules: The model outperforms previous local-learning methods on unsupervised representations.
- Continual-learning results: It surpasses competing approaches on Split-CIFAR/Tiny-ImageNet.
- learns representation that are transferable across image datasets
- Impressive reconstructions: The decoder shows that most image features are preserved in the latent space.

Originality

- Novel local learning rule: The proposal is clearly distinct from earlier Hebbian or STDP variants.
- Auxiliary projection + similarity loss: This combination extends local unsupervised learning by mapping the latent state back to its input without enforcing constraints that might be too prohibitive (L2 loss) and allowing for different projection dimensions.

## Weaknesses

Quality

- Not fully Hebbian: The porposed learning method is not fully hebbian for three reasons. 1) Non-local in time: The off-diagonal component of $ XX^\top$ and $ YY^\top$ are not hebbian because they couple activities from different time steps (batch samples). 2) Non-local in space: the off-diagonal components of $ Y^\top Y$ require access to all neurons in $Y$. 3) Weight updates in the main network depend on gradients flowing through the auxiliary projection, violating strict locality. While items 1 and 2 might be approximated with online updates and/or lateral inhibition, point 3 is harder to reconcile with Hebbian principles.
- Transfer-learning baseline missing: Without a reference model it is hard to gauge the significance of the reported gains.

Clarity

- Evaluation method not present in the main text: Section 5.1 should clearly state that classification accuracy is obtained by training the fully connected output layer to classify image labels from the representations learned by the model.
- No algorithm box: A concise pseudo-code block that shows gradient detachment to prevent full back-prop would improve reproducibility.

Significance

- Speculative explanation of continual-learning performance: The paper offers a hypothesis but no empirical evidence.
- Back-prop comparisons lacking: Since the method partly relies on back-propagation, it should also be compared with back-prop-based unsupervised learners in §5.2 because fully Hebian methods are at a disadvantage compared to the proposed method
- Opaque reconstruction evidence: Table 6 in the supplementary should appear in the main text to support the reconstruction claims.

Originality

- Orthogonal-representation losses are not new: Several prior works have already explored orthogonality constraints.

---

> ### Author Rebuttal · Authors · 2025-07-30
>
> **Comment 1:**
>
> Not fully Hebbian: ...
>
> **Response to comment 1:**
>
> We are grateful to the reviewer for this insightful critique. We agree that SPHeRe is not a literal, synaptic-level implementation of a classical Hebbian rule, but rather a computational model that abstracts its core functional principles. Regarding non-locality in time and space (points 1 & 2), we view batch-level computations ($XX^\top, YY^\top$) as a computational abstraction for estimating input statistics over a short time window, and the orthogonality constraint ($L_{orth}$) as a functional analog of lateral inhibition, which promotes competition and decorrelation. We acknowledge that point 3—gradients flowing through the auxiliary module $\phi$—is the most significant departure. However, we interpret $\phi$ not as a non-local feedback path like in backpropagation, but as a functional abstraction of local error-generating circuits, such as dendritic compartments. These biological microcircuits can compute local prediction errors to guide plasticity within the same block, without requiring symmetric weights or global signals. This feedforward "main pathway learning, side-pathway supervision" architecture is a deliberate design choice that allows us to apply Hebbian-inspired principles to complex, non-linear deep networks. We will clarify this "functional abstraction" perspective in the revised manuscript.
>
>
> **Comment 2:**
>
> Transfer-learning baseline missing: Without a reference model it is hard to gauge the significance of the reported gains.
>
> **Response to comment 2:**
>
> We thank the reviewer for this valuable suggestion. We agree that a reference model is crucial for contextualizing our transfer learning results. To address this, we will conduct additional experiments using SoftHebb as a direct baseline, as it is the state-of-the-art method we compare against in our main classification tasks. We will follow the identical protocol: pre-training SoftHebb on the source dataset (e.g., Tiny-ImageNet) and then evaluating the frozen features on the target dataset (e.g., CIFAR-10). The results will be added to Table 4 in the revised manuscript, providing a direct comparison that will better highlight the significance of SPHeRe's transferability. For convenience, we also display the complete table below.
>
>  ```
> | Method   | Transfer Direction         | Transfer Learning (%) (Train/Test) | Non-Transfer (%) | Gap (%)   |
> | -------- | -------------------------- | ---------------------------------- | ---------------- | --------- |
> | BP       | Tiny-ImageNet → CIFAR-10   | 100 / 81.7                         | 88.7             | -7.0      |
> | BP       | CIFAR-10 → Tiny-ImageNet   | 99.9 / 39.2                        | 45.3             | -6.1      |
> | SoftHebb | Tiny-ImageNet → CIFAR-10   | 78 / 74.84                         | 78.86            | −4.02     |
> | SoftHebb | CIFAR-10 → Tiny-ImageNet   | 60 /33.26                          | 34.12            | -0.86     |
> | SPHeRe   | Tiny-ImageNet → CIFAR-10   | 97.3 / 80.03                       | 81.11            | −1.08     |
> | SPHeRe   | CIFAR-10 → Tiny-ImageNet   | 99.9 / 37.7                        | 40.33            | −2.63     |
>  ```
>
> **Comment 3:**
>
> Evaluation method not present in the main text: Section 5.1 should clearly state that classification accuracy is obtained by training the fully connected output layer to classify image labels from the representations learned by the model.
>
> **Response to comment 3:**
>
> The reviewer is correct, this information is crucial for reproducibility. Although we mentioned that the model includes "a fully connected output layer trained with backpropagation" in Section 5.1, we agree that this was not explicitly framed as the evaluation method. In the revised manuscript, we will clarify the training process more explicitly.
>
> **Comment 4:**
>
> No algorithm box: A concise pseudo-code block that shows gradient detachment to prevent full back-prop would improve reproducibility.
>
> **Response to comment 4:**
>
> Thanks for this suggestion. We agree that a pseudo-code algorithm box is essential for clarity and reproducibility. In the revised manuscript, we will add an algorithm box detailing the training process.
>
> **Comment 5:**
>
> Speculative explanation of continual-learning performance: The paper offers a hypothesis but no empirical evidence.
> Please explain why you expect the model not to suffer from catastrophic forgetting. An ablation that isolates each loss term’s contribution could help identify the mechanism.
>
> **Response to comment 5:**
>
> We thank the reviewer for this crucial point. We speculate that the robustness of SPHeRe in continual learning mainly stems from our loss function, which encourages task-agnostic representations. $L_{SPHeRe}$ focuses on preserving the fundamental structure of the input data itself, rather than task-specific discriminative features, while $L_{orth}$ promotes decorrelation in the feature space, which may reduce interference between features learned for different tasks. We conduct ablation experiments on the Split-CIFAR100 benchmark, comparing the continual learning performance of the complete SPHeRe method (73.95%) with a version that uses only the $L_{SPHeRe}$ loss (73.8%).
>
> **Comment 6:**
>
> Back-prop comparisons lacking: Since the method partly relies on back-propagation, it should also be compared with back-prop-based unsupervised learners in §5.2 because fully Hebian methods are at a disadvantage compared to the proposed method
>
> **Response to comment 6:**
>
> This is a very insightful point, and we agree that comparison with the backpropagation-based method is important. However, we think that a direct performance comparison with mature unsupervised learning (e.g., a contrastive learning method like SimCLR), even in a block-wise setting, requires careful contextualization, as it highlights a fundamental difference in learning philosophy. Current SOTA unsupervised methods rely on a strong, explicit pseudo-supervisory signal derived from sophisticated data augmentations. This is a powerful, human-designed heuristic for learning discriminative features. In contrast, SPHeRe runs on a different principle, it uses a simple and unsupervised objective that aims to preserve the local information structure of the input, without relying on such explicit supervisory signals. We provided a comparison with a self-supervised learning method, but for the reasons mentioned above, we do not believe that the wins or losses in comparison to it will diminish the contribution of our method. We use SimCLR to train the same network for 100 epochs, and achieve the test accuracy 71.94%. Although the accuracy of SimCLR is lower than that of SPHeRe, this does not mean that the SimCLR method is inferior; rather, it indicates that the two methods are suitable for different model structures. SPHeRe is suitable for shallower but wider networks, while SimCLR is suited for narrower and deeper networks.
>
> **Comment 7:**
>
> Opaque reconstruction evidence: Table 6 in the supplementary should appear in the main text to support the reconstruction claims.
>
> **Response to comment 7:**
>
> We thank the reviewer for this valuable comment. In the revised manuscript, we will move the quantitative results from Table 6 into the main experimental section.
>
> **Comment 8:**
>
> Orthogonal-representation losses are not new: Several prior works have already explored orthogonality constraints.
>
> **Response to comment 8:**
>
> We agree that orthogonality constraints are an established technique, and our novelty does not lie in the invention of the loss itself. Rather, our contribution is the specific application and motivation. In this paper, it works with $L_{SPHeRe}$ to limit the output features of each dimension to be as orthogonal as possible. In practical use, L_orth is applied to the low-dimensional auxiliary features Z, rather than the main high-dimensional output Y, which avoids the high computational cost of high-dimensional features $Y^\top Y$ and improves computational efficiency. At the same time, the learning process implicitly encourages the main block f to generate features Y that are easier to be separated and projected onto an orthogonal basis by a simple $\phi$ network, encouraging Y to become more decorrelated and less redundant. In the revised manuscript, we will add corresponding explanations to clarify our application of $L_{orth}$.
>
> **Comment 9:**
>
> Lemma 4.1 assumes that $M < N$ . This is not true in many model architectures researchers might consider (and in the brain, eg generative mdoels). How does the model performance change if $M\approx N$   or $M > N$ ?
>
> **Response to comment 9:**
>
> Thank you to the reviewers for pointing out the limitations of the lemma. We acknowledge that the restriction $M < N$ is intended to demonstrate that our objective function can effectively preserve the main information when compressing data. When $M\approx N$ and $M > N$, $L_{SPHeRe}$ does not have a unique solution, so at this point, the orthogonal loss$ L_{orth}$ becomes particularly important, as it prevents the model from learning simple identity transformations or redundant feature spaces. Once we introduce the auxiliary mapping $\phi$, the relationship constraints between M and N will be unlocked, and the output Y will always be able to effectively encode the main features of X.

---

> > ### Comment · Reviewer_dLRw · 2025-08-04
> > **Disadvantage of fully Hebian methods**
> >
> > Thank you for addressing the comments. I especially appreciated the clarification on transfer learning which was very clear.
> >
> > Could you please further clarify Comment 6:
> > From reading the SoftHebb paper it seems like it is fully local and has fewer parameters than your method (it does not have auxilary projection). Could you further clarify how the partial integration of backpropagation in your method constitutes a fair and meaningful comparison with fully local models like SoftHebb?

---

> ### Author Response · Authors · 2025-08-05
>
> Thank you for this insightful and important question. We agree that SPHeRe and SoftHebb represent two different paths for integrating Hebbian principles into modern deep networks. SoftHebb successfully applies Hebbian rules to multi-layer convolutional networks by introducing a Soft-WTA mechanism, whereas SPHeRe employs a functional abstraction of the core Hebbian principle (preserving input structure) via an auxiliary module $\phi$. However, we would argue that characterizing SoftHebb as "fully local" might overlook some key compromises made to achieve high performance in a modern framework. Our analysis indicates that SoftHebb also deviates from the strictest biological constraints since there exists implicit cross-sample information aggregation in its code: In its weight update $\Delta W$ calculation, the implementation of the $yx$ term cleverly but critically uses information from the entire mini-batch to compute a single, unified weight update. This differs from strict biological locality, where an update depends only on a single sample.
>
> Therefore, we believe the comparison between SPHeRe and SoftHebb is not a contest between a "purely local" model and a "hybrid BP" model. Instead, it is a comparison of two different "abstraction strategies" for effectively combining the essence of Hebbian principles with modern deep learning. SPHeRe's strategy is conceptually closer to the optimization paradigm of modern deep learning and demonstrates superior practicality and performance.
>
> Compared to SoftHebb, SPHeRe, as an objective-driven learning framework, offers several advantages:
>
> 1. Theoretical Soundness and Interpretability: SPHeRe is founded on a clear optimization objective ($L_{SPHeRe}$), and its core idea—preserving the geometric structure of input data—is supported by solid mathematical justification. This makes our method principled, interpretable, and easy to analyze and extend.
>
> 2. Generality and Flexibility: The SPHeRe framework is less dependent on specific network components. As shown in our experiments (Table 7 and 8), it is compatible with various standard activation functions and can even be adapted to Spiking Neural Networks (SNNs). SoftHebb, on the other hand, relies  on its custom Triangle activation function, which limits its generality.
>
> 3. Ease of Use and Robustness: SPHeRe's training is more stable and less sensitive to hyperparameters, allowing for a single learning rate across all layers. In contrast, SoftHebb requires manually tuning different learning rates and several other hyperparameters (t_invert, power) for each layer, which increases the complexity and potential arbitrariness of the tuning process.
>
> To further validate that SPHeRe’s performance doesn’t merely come from the increased parameter count, we conducted an ablation study. First, we increased the number of channels of the SoftHebb network structure to match SPHeRe’s parameter count (an approximate 15% increase by adding the auxiliary modules). This modification, however, resulted in a slight decrease in accuracy on the CIFAR10 test set, from 78.86% to 78.63%. This situation indicates that simply widening the network does not yield performance benefits when using Softhebb. Then, we reduce the number of channels in the SPHeRe model by one-third (making the parameter count significantly smaller than the SoftHebb situation), and its test accuracy remains remarkably high at 80.7%, a small drop of only 0.4% from the original model’s 81.11%. These results provide evidence that SPHeRe’s superior performance originates from its learning framework and optimization objective, rather than the parameter scale.
>
> In summary, SPHeRe trades a modest parameter overhead for comprehensive improvements in performance, versatility, and robustness—a trade-off we believe is highly effective. More importantly, it charts a clear path for bio-inspired learning to evolve from procedural rules toward objective-driven optimization, showcasing its immense potential within modern deep learning frameworks.

---

### Official Review · Reviewer_RoRs · 2025-06-27

**Clarity:** 3
**Significance:** 4
**Originality:** 4
**Rating:** 5
**Confidence:** 5

**Summary:**

This paper rethinks the Hebbian learning rule and proposes an improved unsupervised learning method called SPHeRe. SPHeRe simplifies the equivalent loss function of the Hebbian OJa’s rule and introduces auxiliary blocks and orthogonality constraints. The paper has proved that optimizing the proposed loss function is equivalent to finding the best low-dimensional projection of the input data. SPHeRe achieves the current best classification accuracy among unsupervised synaptic plasticity methods and performs well in continual learning and transfer learning.

**Questions:**

1.The target of the orthogonal loss in the paper is applied to the auxiliary feature Z. Is this effective for the high-dimensional feature Y?
2.Can the SPHeRe method be combined with self-supervised learning to achieve better results?

**Ethical Concerns:**

["NO or VERY MINOR ethics concerns only"]

**Final Justification:**

Thanks for response.

**Limitations:**

Yes.

**Paper Formatting Concerns:**

No.

**Quality:**

3

**Strengths And Weaknesses:**

Strength
The paper build a bridge between the Hebbian rule and a clear and optimizable loss function. The proposed SPHeRe loss function is simple but has explicit mathematical significance (seeking the optimal low-dimensional projection). This provides a solid theoretical foundation and development perspective for Hebbian-like rules that originally lacked clear optimization objectives.

Weakness
1.The method proposed in the paper reduces the locality constraints of Hebbian-like rules, where its learning rule depends on the introduced auxiliary blocks for gradient backpropagation to update the backbone network layers. Although SPHeRe is trained layer by layer on the backbone network, it actually uses backpropagation at the block level.
2.The core theory of the paper is derived from linear transformations, but it has been extended to nonlinear networks. The authors argue that due to the universal approximation ability of neural networks, they have the capacity to learn the theoretically optimal solution. However, even though the authors have empirical validated this through experiments in the appendix, it does not constitute a strict theoretical guarantee. Moreover, the introduction of the auxiliary network has improved final performance, possibly allowing the network to learn more powerful features.
3.Table 2 shows that using L_{SPHeRe} alone is less effective than the original L_{oja} loss, and the paper does not adequately discuss and explain this phenomenon.
4.The ablation experiments in the paper demonstrate the effectiveness of the auxiliary block, but the authors have not analyze in depth how its structure affects the results.
5.In the caption of Figure 2, “10^3” should be “10^{-3}”.

---

> ### Author Rebuttal · Authors · 2025-07-30
>
> **Comment 1:**
>
> The method proposed in the paper reduces the locality constraints of Hebbian-like rules, where its learning rule depends on the introduced auxiliary blocks for gradient backpropagation to update the backbone network layers. Although SPHeRe is trained layer by layer on the backbone network, it actually uses backpropagation at the block level.
>
> **Response to comment 1:**
>
> We thank the reviewer for this precise observation, which is entirely correct. SPHeRe indeed uses backpropagation for credit assignment within each block, and its claim to locality is at the block level, not the individual synaptic level. This design is a deliberate and pragmatic choice. Our primary goal is to apply Hebbian principles—specifically, preserving input-output structural similarity—to train modern, deep, non-linear networks effectively. While classical Hebbian rules are synaptically local, they lack a clear mechanism for credit assignment within a complex, multi-layer computational block (like a ResNet block). Our solution is to define a Hebbian-inspired local objective for the block (comparing its input X to its projected output Z) and then use backpropagation as an efficient and well-established computational tool to optimize the block's parameters to meet that objective. The auxiliary block and intra-block backpropagation function together as a computational abstraction for a local error circuit (e.g., dendritic computation), which guides plasticity based on information available locally to the module.
>
> **Comment 2:**
>
> The core theory of the paper is derived from linear transformations, but it has been extended to nonlinear networks. The authors argue that due to the universal approximation ability of neural networks, they have the capacity to learn the theoretically optimal solution. However, even though the authors have empirical validated this through experiments in the appendix, it does not constitute a strict theoretical guarantee. Moreover, the introduction of the auxiliary network has improved final performance, possibly allowing the network to learn more powerful features.
>
> **Response to comment 2:**
>
> We fully acknowledge that our theoretical analysis is grounded in a linear setting and that extending it to deep, nonlinear networks relies on the universal approximation capacity, which, as the reviewer correctly states, is not a formal guarantee of convergence to the global optimum. Our justification is two-fold. First, as we argue in our response, the purpose of Lemma 4.1 is to provide a strong theoretical motivation for the $L_{SPHeRe}$ loss. It demonstrates that the objective $||X^\top X - Y^\top Y||^2_F$ is principled and has a clear, desirable target in an analyzable setting: finding the optimal principal subspace. This provides a solid foundation for its use, moving it beyond a mere heuristic. Second, the key to applying this objective to deep networks is our proposed purely feedforward, decoupled architecture. We introduce a lightweight auxiliary module ($\phi$) that operates alongside the main backbone block ($f$). This "main-pathway learning, side-pathway supervision" design creates a crucial separation of concerns: 1) The main block (f) is not forced to learn a direct, low-dimensional projection of the input. Instead, it can learn richer, higher-dimensional features (Y') that are more powerful for downstream tasks; 2) The auxiliary network $\phi$ then bears the responsibility of projecting these rich features into a low-dimensional space Z where the structural preservation loss is computed. The empirical results in Appendix B, showing high CKA similarity, serve as strong evidence that this decoupling strategy successfully guides the nonlinear system to learn representations that are functionally equivalent to the theoretical linear optimum, even if a formal proof remains an open challenge.
>
> **Comment 3:**
>
> Table 2 shows that using L_{SPHeRe} alone is less effective than the original L_{oja} loss, and the paper does not adequately discuss and explain this phenomenon.
>
> **Response to comment 3:**
>
> We thank the reviewer for this keen observation. It is a crucial point that highlights the core design trade-off behind SPHeRe. While $L_{oja}$ appears superior in isolation, its strength comes from the $(XX^\top)^{⁻¹}$ term, which acts as an adaptive weighting mechanism. This can be powerful, but it also introduces significant numerical instability and high computational cost, especially in deep networks where the input feature maps can be high-dimensional and ill-conditioned, making the matrix inversion impractical or unreliable. By removing this inverse term, we consciously traded some of $L_{oja}$'s isolated adaptive power for a stable foundation. The true value of$ L_{SPHeRe}$ is revealed not in isolation, but in its synergy with the other components of our method. As Table 2 shows, the simplified and stable nature of$ L_{SPHeRe}$ allows it to be effectively combined with the orthogonality constraint ($L_{orth}$) and the auxiliary block ($\phi$), leading to the overall best performance. The brittleness of $L_{oja}$ makes such stable integration more challenging. We will explicitly add this discussion to the revised manuscript to clarify this important design choice.
>
> **Comment 4:**
>
> The ablation experiments in the paper demonstrate the effectiveness of the auxiliary block, but the authors have not analyze in depth how its structure affects the results.
>
> **Response to comment 4:**
>
> In the manuscript, the auxiliary block $\phi$ adopts the simplest nonlinear network, which consists of only one convolutional layer with a kernel size of 1, an average pooling layer that downsamples the feature map to 1, and a fully connected projection layer. Here we provide the results after changing the convolutional layer and the dimension of the projection layer.
>  ```
> | convolutional layers      | 0     | 1     | 2     |
> | ------------------------- | ----- | ----- | ----- |
> | CIFAR10 Test accuracy (%) | 78.65 | 81.11 | 79.93 |
>  ```
>
>  ```
> | Project dimension         | 128  | 256   | 512   |
> | ------------------------- | ---- | ----- | ----- |
> | CIFAR10 Test accuracy (%) | 80.5 | 81.11 | 81.18 |
>  ```
>
> **Comment 5:**
>
> In the caption of Figure 2, “10^3” should be “10^{-3}”.
>
> **Response to comment 5:**
>
> We sincerely thank the reviewer for their meticulous reading and for catching this typo, we will correct it in the revised manuscript.
>
> **Comment 6:**
>
> The target of the orthogonal loss in the paper is applied to the auxiliary feature Z. Is this effective for the high-dimensional feature Y?
>
> **Response to comment 6:**
>
> Thank you for this excellent question, which points to a crucial implementation detail that we admit was not perfectly clear in the manuscript.  To clarify, in our final implementation, the orthogonality loss $L_{orth}$ is indeed applied to the low-dimensional auxiliary feature Z for computational efficiency, as this avoids the high cost of computing $Y^\top Y$ for a high-dimensional feature Y. The mechanism is one of indirect regularization. By optimizing the parameters of both the main block ($f$) and the auxiliary block ($\phi$) to make $Z=\phi(Y)$ orthogonal, the learning process implicitly forces the main block f to produce features Y that are more easily separable and projectable into an orthogonal basis by the simple $\phi$ network. In essence, this encourages Y to become more decorrelated and less redundant, acting as a computationally tractable proxy for promoting good feature separation in the high-dimensional space. We will revise the manuscript to resolve any inconsistency and explicitly describe this mechanism.
>
> **Comment 7:**
>
> Can the SPHeRe method be combined with self-supervised learning to achieve better results?
>
> **Response to comment 7:**
>
> We thank the reviewer for this highly insightful question. The combination of SPHeRe and self-supervised learning is indeed an interesting direction. We guess that combining our local, block-level self-supervision with a global, cross-sample SSL objective (e.g., SimCLR or BYOL) could create a powerful, multi-scale representation learning method, but this represents an exciting direction for future work.

---

> > ### Comment · Reviewer_RoRs · 2025-08-04
> > **The explanations you've provided have effectively resolved the questions I raised earlier.**
> >
> > Thank you for your detailed response. The explanations you've provided have effectively resolved the questions I raised earlier, and the work now appears much clearer to me. I appreciate the time and effort you've put into clarifying these aspects.

---

### Official Review · Reviewer_b3Ah · 2025-07-01

**Clarity:** 3
**Significance:** 4
**Originality:** 3
**Rating:** 5
**Confidence:** 5

**Summary:**

This paper provides a similarity-based representation learning approach. This similarity measure allows the 'inputs' of a module to be nonlinearly transformed before applying the similarity measure. The authors also add some orthogonality constraints on intermediate-layer neurons. This approach with multiple similarity blocks/layers generates representations that are good for supervised learning, but also do well for continual learning and transfer learning. The representations learned also could be used for an auto-encoder.

**Questions:**

Clarification of how the labels were used in their evaluation would help a broader set of readers, some of whom may come from computational neuroscience.

Contrasting with previous work would place the work in the proper context. In that vein, could the authors comment on how their training relates to Hebbian learning as understood in neurobiology?

**Ethical Concerns:**

["NO or VERY MINOR ethics concerns only"]

**Final Justification:**

I believe the authors have tried to change their manuscript and responded to the reviews constructively. I think this work is an interesting starting point for multilayer similarity-based representation learning, even if it is not yet fully Hebbian. I don't know whether some day we will get deep models with hierarchical feature learning, with Hebbian rules, or not. However, it as a serious effort in that direction. So, I am upgrading my score.

**Limitations:**

Yes.

**Paper Formatting Concerns:**

None.

**Quality:**

3

**Strengths And Weaknesses:**

Finding a similarity-based multilayer network where different layers learn more abstract features has been a goal for many researchers, especially those interested in biologically plausible networks. It seems that the authors have made some progress there. Compared to some other unsupervised methods inspired by Hebbian learning, the authors claim that they have better performance.

One major complaint I have is that the paper is written as if there has been not much work previously done on similarity-based objectives and Hebbian learning. It completely ignores works like Pehlevan and Chklovskii, Asilomar, 2014, Pehlevan, Hu, and Chklovskii, Neural Computation, 2015; Pehlevan and Chklovskii, NeurIPS, 2015; Pehlevan, Sengupta and Chklovskii, Neural Computation, 2017; Obeid, Ramambason, and Pehlevan, NeurIPS, 2019. Some of these previous works were mentioned in the Nokland and Eidnes paper the authors cite. Contrasting with these works gives a better appreciation of authors' innovations. Also, some of the references to arXiv papers should be changed to their published version. For example, Saha et al is in ICLR 2021 and the Journe et al paper is in ICLR 2023.

It might also be helpful to be explicit about how the labels are being used. My impression from reading the paper is that representation learning does not use labels at all, and I imagine only later some shallow supervised learning is done using the learned representation. Some similarity-based methods like HSIC Bottleneck (Ma et al, AAAI, 2020; Pogodin and Latham, NeurIPS, 2020), use both inputs and labels while training networks. Hence a clear description of the method may help.

---

> ### Author Rebuttal · Authors · 2025-07-30
>
> **Comment 1:**
>
> One major complaint I have is that the paper is written as if there has been not much work previously done on similarity-based objectives and Hebbian learning. It completely ignores works like Pehlevan and Chklovskii, Asilomar, 2014, Pehlevan, Hu, and Chklovskii, Neural Computation, 2015; Pehlevan and Chklovskii, NeurIPS, 2015; Pehlevan, Sengupta and Chklovskii, Neural Computation, 2017; Obeid, Ramambason, and Pehlevan, NeurIPS, 2019. Some of these previous works were mentioned in the Nokland and Eidnes paper the authors cite. Contrasting with these works gives a better appreciation of authors' innovations. Also, some of the references to arXiv papers should be changed to their published version. For example, Saha et al is in ICLR 2021 and the Journe et al paper is in ICLR 2023.
>
> **Response to comment 1:**
>
> We are very grateful to the reviewer for pointing out the series of foundational works by Pehlevan, Chklovskii, and their collaborators. We fully agree that placing our work in this important academic context for discussion is the key to enhancing the quality of the paper. Our work shares a similar mathematical starting point with these foundational works: optimizing a similarity matching objective of the form $||X^\top X - Y^\top Y||^2_F$. However, the path we take to solve the core challenge of "how to effectively apply this objective to deep nonlinear networks" is distinctly different, which precisely highlights the unique contribution of our SPHeRe method.
>
> Pehlevan, Chklovskii, Obeid (2019), et al.'s path: They elegantly derived local learning rules with Hebbian/anti-Hebbian forms starting from similarity matching objectives through methods such as min-max dual optimization and extended it to deep networks. As described by Obeid et al., their deep model theoretically relies on feedback connections to achieve credit assignment across layers.
>
> Our path: We start from the equivalent loss function of the Hebbian rule (specifically Oja's rule) and simplify it to achieve the same objective. However, to apply it to deep networks, we propose a different, purely feedforward architecture. We introduce a decoupled lightweight auxiliary module ($\phi$) that projects the high-dimensional features of the backbone network into a low-dimensional space and calculates similarity loss in this low-dimensional space. This design of "main-pathway learning, side-pathway supervision" structurally avoids the need for explicit feedback paths, making the model more modular and seamlessly integrable into mainstream deep learning frameworks.
>
> In the revised manuscript, we will clearly articulate this "different paths lead to the same goal" derivation process in the methodology section and compare in depth the differences between our feedforward block-level training scheme and feedback-dependent schemes to more accurately position our work. Meanwhile, we will re-examine the issues regarding the version of the cited papers you mentioned in the revised manuscript.
>
>
> **Comment 2:**
>
> Clarification of how the labels were used in their evaluation would help a broader set of readers, some of whom may come from computational neuroscience.
>
> **Response to comment 2:**
>
> Thank you for your insightful feedback, and we agree that clarifying our evaluation methodology is crucial for a broad readership, particularly those with a background in computational neuroscience. We apologize if our initial description was not sufficiently clear.  To clarify, the core of our method, SPHeRe, is entirely unsupervised. The feature learning process in the convolutional layers does not use any ground-truth labels. The training of these layers relies solely on the SPHeRe loss, which aims to preserve the structural information from the input data itself, aligning with biologically plausible, local learning rules. We added a fully connected layer as a classifier after the convolutional layers, and only this last linear layer is trained in a supervised manner, using the standard cross-entropy loss function and image labels. We will revise the experimental setup in our manuscript to explicitly and clearly describe this two-stage training and evaluation process to prevent any ambiguity.
>
> **Comment 3:**
>
> Contrasting with previous work would place the work in the proper context. In that vein, could the authors comment on how their training relates to Hebbian learning as understood in neurobiology?
>
>
> **Response to comment 3:**
>
> Thank you for this insightful question. We agree that this is a crucial point. We position SPHeRe not as a literal, synaptic-level copy of neural circuits, but as a functional and computational model that abstracts and implements key principles of biological learning. Instead of viewing plausibility as a binary "yes/no" state, we see our work as a step forward from end-to-end backpropagation. At its core, our learning rule retains the fundamental associativity of Hebbian learning ("fire together, wire together"). However, our main contribution lies in addressing the inherent instability of classical Hebbian learning by incorporating computational analogs of more advanced neural regulation mechanisms:
>
> Our core loss, $L_{SPHeRe}$​, is an implementation of the Efficient Coding Hypothesis. As our lemma shows, in the linear case, this objective is equivalent to PCA, a classic redundancy reduction strategy. Furthermore, based on the pioneering work of Pehlevan and Chklovskii et al., this objective can also be understood as a predictive similarity matching, and the entire optimization process can be decomposed into fully local Hebbian and anti-Hebbian update rules at each synapse.
>
> The orthogonality constraint, $L_{orth}$​, computationally mirrors the function of lateral inhibition, promoting competition among neurons to learn efficient and decorrelated features.
>
> The auxiliary module ($\phi$) serves as an abstraction of local error circuits, such as dendritic compartments, which are theorized to guide synaptic plasticity locally without requiring the symmetric weights or global error signals that challenge standard backpropagation.
>
> We acknowledge that batch-level computations are a computational abstraction. We view a "batch" as a set of stimuli arriving in a short time window, where the computation estimates the input statistics that a neural circuit might capture through biophysical means.
>
> In summary, our goal is to model the functional objectives of biological learning, and we believe SPHeRe makes a significant step towards biological plausibility by modeling the functional goals—associativity, homeostasis, and competition—of neural learning systems.

---

> > ### Comment · Reviewer_b3Ah · 2025-08-06
> >
> > Thank you for detailed and careful responses. I appreciate it very much!

---

### Note · Authors · 2025-08-12

Dear Area Chair and Reviewers,

We are profoundly grateful for your insightful and highly constructive feedback throughout the review process. Your guidance has significantly enhanced the quality and rigor of our work. During the rebuttal, we diligently worked to address the major concerns raised, implementing several key improvements and conducting new experiments based on your suggestions:

1. Refined Positioning & Contribution Clarity: We have adopted your advice to reposition our paper as a "Hebbian-inspired, greedy layer-wise unsupervised pre-training method." We clarified that "biological plausibility" is framed as a functional abstraction to avoid overclaiming. We also detailed the distinction from foundational works like Pehlevan et al., highlighting our method's unique implementation path.

2. New Crucial Experiments for Stronger Support: \
• To address scalability concerns, we added a ResNet-18 stem pre-training experiment. The results demonstrate SPHeRe's practical potential as a pre-training module for modern architectures (79.91% vs. 79.46% on CIFAR-100). \
• For a fair performance comparison, we included benchmarks against greedy layer-wise Auto-Encoders and SimCLR, validating SPHeRe's competitiveness within this paradigm (SPHeRe 81.18% vs. AE 79.26% and SimCLR 75.07%). \
• Furthermore, we added transfer learning baselines and parameter-count ablations to solidify our method's advantages.

We commit to fully integrating all the above discussions, new experimental results, and promised revisions (including pseudocode, citations, and typo corrections) into the final version.

We sincerely appreciate the reviewers' prompt responses and their final positive acknowledgements. Thank you again for your invaluable time and expertise, which have elevated our work to a new level.

---

### Decision · Program_Chairs · 2025-09-17

**Decision:**

Accept (poster)

**Comment:**

Inspired by Oja's Hebbian learning rule, this work proposes a similarity-based objective for unsupervised representation learning. They optimize their objective in a deep neural network using the backpropagation algorithm and show that it performs favorable when compared with other unsupervised learning objectives/algorithms. Multiple reviewers took issue with two points: (a) the paper was misleading and led the reader to believe that the authors were proposing "biologically plausible" learning algorithm; (b) that the work did not situate itself within a rather extensive existing literature on similarity-based objectives that have been used to derive "biologically plausible" learning algorithms. There were also some concerns about the benchmarks for comparison with existing unsupervised models. In the rebuttal and during the discussion period, the authors clarified that their work is "biologically inspired" rather than "biologically plausible", promised to properly place their work within the existing literature on similarity-preserving objectives and performed additional experiments. After the discussion period, several reviewers raised their scores. In the end, some of the reviewers still thought the paper could use another round of revisions, while others thought that the authors had adequately addressed their concerns. Ultimately, I think that the main objections from the reviewers can be addressed by the authors in their revision and so I recommend acceptance.